# The Role of Toss Zenith and Impact Height in the Relationship Between Shoulder Rotation Strength and Serve Speed in Junior Tennis Players

**DOI:** 10.3390/jfmk10040438

**Published:** 2025-11-10

**Authors:** Jan Vacek, Michal Vagner, Jan Malecek, Jachym Simsa, Vladan Olah, Petr Stastny

**Affiliations:** 1Department of Sports Games, Faculty of Physical Education and Sport, Charles University in Prague, 162 52 Prague, Czech Republic; petr.stastny@ftvs.cuni.cz; 2Department of Military Physical Education, Faculty of Physical Education and Sport, Charles University in Prague, 162 52 Prague, Czech Republic; jan.malecek@ftvs.cuni.cz (J.M.); simsa.ftvs@gmail.com (J.S.); vladan.olah@ftvs.cuni.cz (V.O.)

**Keywords:** toss kinematics, biomechanics, isokinetic strength

## Abstract

**Background:** Serve speed in tennis can be influenced by physical strength and toss kinematics, yet their interaction remains unclear in junior athletes. This study investigated the relationships between toss-related variables (zenith height, impact height, and zenith–impact differential), shoulder rotation strength, and serve speed in junior tennis players. **Methods:** Thirteen nationally ranked junior male players (mean age: 15.8 ± 0.9 years; body mass: 65.9 ± 5 kg; height: 181 ± 7 cm) participated. Isokinetic shoulder internal and external rotation strength (concentric/eccentric at 210°/s and 300°/s) were measured. High-speed 3D motion capture (Qualisys) recorded toss zenith, impact height and zenith–impact differential during flat serves directed into a designated 1 × 2 m target zone. Serve speed was measured by a radar gun. Correlation, mediation, and moderation analyses were conducted (*n* = 13; sensitivity ρ ≥ 0.69 for 80% power). **Results:** Impact height showed a significant positive correlation with successful (ρ = 0.746, *p* = 0.003) and unsuccessful serve speed (ρ = 0.776, *p* = 0.002). Moderation analyses showed that lower variability in impact height (*p* = 0.043) and in zenith–impact differential (*p* = 0.048) significantly strengthened the association between shoulder-rotator strength and serve speed. **Conclusions:** Moderation analyses revealed that lower variability in impact height and in the zenith–impact differential strengthened the relationship between shoulder strength and serve speed. Preliminary evidence suggests that consistent toss execution could enhance the translation of shoulder-rotation strength into serve speed in junior players.

## 1. Introduction

The tennis serve plays a decisive role in competitive play, strongly influencing point initiation and match outcomes [1,2,3]. Its effectiveness depends primarily on serve speed and placement, which determine an opponent’s ability to return the ball successfully [4,5,6,7,8]. High serve velocities rely on an efficient kinetic chain that sequentially transfers energy from the lower limbs through the trunk to the upper extremity, and racket head [9,10]. Disruptions in this coordinated sequence, such as inconsistencies in ball toss execution, may require compensatory adjustments that reduce power transfer and accuracy [11,12,13,14]. Therefore, identifying biomechanical and neuromuscular factors that support efficient kinetic sequencing is essential for improving serve performance and developing evidence-based training strategies.

A stable and precise ball toss is essential for serve consistency, as it directly affects both power generation and shot accuracy [15,16,17]. Key biomechanical characteristics, including the initial toss position, toss height, and the alignment between the ball’s apex and its impact point, determine whether the player can achieve an optimal impact point that maximizes timing and kinetic efficiency during ball–racket interaction [5,16]. Excessive variability in toss placement may disrupt the kinetic sequence, requiring compensatory movements at the shoulder, elbow, or wrist that reduce energy transfer and increase the likelihood of unforced errors [17]. These issues are often more pronounced in junior players, who typically exhibit greater toss variability due to developing motor control and coordination [18,19,20]. Understanding which toss-related metrics (e.g., zenith height, impact height, and their differential) most strongly relate to serve outcomes may therefore help coaches design targeted interventions that improve technical consistency and facilitate more efficient kinetic-chain transfer.

One key biomechanical indicator in prior research is the vertical distance between the ball’s zenith height and its impact height, known as the zenith–impact differential [21,22,23]. Beyond a geometric description, this parameter reflects how well the descending phase of the toss is synchronized with kinetic-chain activation, particularly trunk rotation and shoulder internal rotation that accelerate the racket [9,10,24,25]. A smaller zenith–impact differential indicates more efficient temporal coordination and thus more effective utilization of shoulder-rotator strength in producing ball speed, whereas a larger differential implies premature or delayed sequencing and compensatory joint actions that dissipate energy [24,25,26]. Existing detailed kinematic data have been collected mainly from professional female players, whose serving mechanics differ from those of developing athletes in trunk–shoulder sequencing and toss consistency [11,19]. Therefore, further investigation is needed in junior population. The ball impact height and zenith–impact differential represent critical variables that may yield deeper technical insights, as toss quality is consistently associated with serve outcomes [25]. The potential mediating role of impact height, specifically its connection to shoulder-rotator strength and overall performance, remains untested. Given that serve performance is also influenced by upper limb strength, particularly shoulder rotator function, examining how the strength parameters interact with kinematic variables is essential.

Previous research has established a significant positive correlation between concentric isokinetic shoulder rotation strength [12,27,28], particularly at 210°/s, and serve speed in junior tennis players [29]. Anthropometric characteristics such as player height, total body mass, and limb length have also been shown to correlate with serve velocity due to biomechanical advantages related to leverage and force production capabilities [9,30]. Specifically, previous findings have indicated that increased upper limb muscle mass significantly facilitates force transfer during the serve motion [29,31]. Building on this foundation, the present study examines how shoulder-rotator strength translates through kinematic parameters, specifically whether impact height mediates this relationship. Clarifying these interdependencies may provide a clearer biomechanical framework linking muscular strength and serve performance, thereby informing more targeted training strategies for junior tennis players.

Finally, serve speed and accuracy are key performance indicators in tennis. High serve speed can effectively limit an opponent’s ability to return [32], whereas accuracy in hitting defined target zones is equally essential for success. Stable impact height has been linked to improvements in both serve speed and accuracy [22,25,32]. From a biomechanical perspective, stronger internal rotators can generate higher racket angular velocities [12,27,28], allowing the player to maintain a more elevated and extended arm position at impact, which may theoretically lead to a higher impact height [9,25]. However, the variability of this contact position may play an even more decisive role: lower variability in impact height or in the zenith–impact differential may allow more efficient translation of shoulder-rotator strength into serve speed [24,25,29,31].

The present study examines the relationships among toss characteristics, specifically zenith height, impact height, and their differential, and serve accuracy, serve speed, and isokinetic shoulder strength variables in junior tennis players. In addition, the study tested whether impact height mediated the relationship between shoulder-rotator strength and serve speed, and whether variability in toss parameters moderated this relationship. Accordingly, two hypotheses were tested: H1 (Mediation): Impact height (M) was expected to mediate the relationship between shoulder internal rotator strength at 210°/s (X) and serve speed (Y). H2 (Moderation): Variability in impact height and in the zenith–impact differential (Moderator) was expected to moderate the strength–speed relationship, such that lower variability would strengthen the positive association between shoulder strength (X) and serve speed (Y).

## 2. Materials and Methods

### 2.1. Study Design

We conducted a cross-sectional observational study extending our prior work [29] by integrating kinematic analysis of the tennis serve with isokinetic shoulder strength testing. The testing protocol comprised two sessions conducted 24 h apart (Figure 1). Session one included isokinetic shoulder strength testing, and Session two comprised kinematic analysis of the serve and serve speed assessment. The study was approved by the Faculty of Physical Education and Sport Ethics Committee at Charles University (No. 243/2020) and conducted in accordance with the Declaration of Helsinki. Written informed consent was obtained from all participants and their legal guardians.

### 2.2. Participants

The study sample comprised thirteen nationally ranked male junior tennis players (mean ± SD: age 15.8 ± 0.9 years; height: 181 ± 7 cm; body mass: 65.9 ± 5 kg; body fat: 17.5 ± 2.2%, eleven right-handed and two left-handed junior tennis players). Body fat percentage was assessed using dual-energy X-ray absorptiometry. The participants in this study were nationally ranked Czech junior tennis players who were actively competing on both national and International Tennis Federation (ITF, ranking 400–800) junior circuits. The sample size (n = 13) was determined by the limited availability of nationally ranked players who met the inclusion criteria and could participate within the constraints of the competitive season. Recruitment was therefore restricted by the national calendar and coordination with coaches, rather than by prior statistical targets. Although the small, homogeneous sample of male Czech juniors limits generalizability, it provided a technically proficient cohort tested under consistent conditions. All players trained approximately five times per week, combining technical, tactical, and physical conditioning sessions at recognized tennis clubs affiliated with national youth sports development programs. Their competitive backgrounds included 8–12 years of training and regular participation in national and ITF junior events. All participants completed one standardized familiarization session with the serve protocol and isokinetic strength protocol prior to testing. Inclusion criteria required participants to be free from performance-limiting injuries or musculoskeletal conditions for at least three months prior to testing.

### 2.3. Testing Protocol

The testing protocol comprised two separate sessions conducted 24 h apart (Figure 1) during the competitive off-season period. Session one involved isokinetic strength testing of the dominant arm using a dynamometer (Humac Norm; CSMi, Stoughton, MA, USA) at two angular velocities: 210°/s and 300°/s. Participants performed three maximal-effort repetitions of internal and external shoulder rotations at each velocity, with standardized rest intervals between tests. The order of testing (angular velocity and internal vs. external rotation) was randomized to minimize order effects. All measurements were conducted under standardized indoor conditions at a controlled temperature of 22 ± 1 °C. Session two consisted of kinematic analysis and measurement of serve speed. Participants completed a standardized 15-min dynamic warm-up (see Section 2.4.2 for details), followed by two sets of 20 maximal-effort flat serves toward a defined target area (1 × 2 m) on the deuce side of the court. Serve speed was measured using a calibrated radar gun (Stalker Pro II, Applied Concepts Inc., Richardson, TX, USA) positioned on the baseline with the ball’s trajectory (≈33 Hz; peak ball velocity in initial free flight). Kinematic data were collected simultaneously using a calibrated six-camera motion capture system (Version 2025.2; Qualisys AB, Gothenburg, Sweden) operating at 200 Hz (5 ms frame interval). Both sessions were designed to ensure participant safety, measurement consistency, and data reliability.

### 2.4. Instruments

This section details the methods and instruments used to measure the isokinetic strength of the shoulder rotators, tennis serve speed, and kinematic variables (zenith height and impact height) in the order in which they were conducted.

#### 2.4.1. Isokinetic Strength Testing of the Shoulder

Isokinetic shoulder rotation strength was assessed using a standard dynamometer (Humac Norm; CSMi, Stoughton, MA, USA) to measure concentric and eccentric internal and external rotations of the dominant (serving) shoulder [33]. Tests were conducted at angular velocities of 210°/s and 300°/s to evaluate peak torque, as these speeds reflect the predominant high-speed muscular contractions characteristic of tennis [27,28,34]. The dynamometer was calibrated at the beginning of each testing day according to the manufacturer’s guidelines to ensure accurate and reliable measurements. Gravity correction was performed prior to each set according to the manufacturer’s guidelines.

Prior to testing, participants completed a comprehensive warm-up to promote muscle readiness and reduce the risk of injury. This included 5 min of general aerobic activity (light jogging with leg swings, arm circles, and trunk rotations), followed by 5 min of shoulder-specific movements (low-resistance internal and external rotations at 90° shoulder abduction, 3 × 10 repetitions each, using elastic resistance bands) to prepare the rotator muscles for high-speed contractions [35]. Participants were positioned supine with the shoulder joint aligned with the dynamometer’s axis of rotation, the arm abducted to 90°, and the elbow flexed at 90°. The shoulder was stabilized with a custom-made humeral support to ensure reproducibility and reduce trunk motion. This testing position provided stability and functional relevance, closely resembling the arm position during the tennis serve [27]. The range of motion was individualized and set to 90% of each participant’s maximal external rotation and 65% of maximal internal rotation. These percentages were determined during a pre-test goniometric assessment of passive shoulder rotation and selected to replicate the functional portion of the serving motion while minimizing capsular stress and impingement risk at end-range positions [27]. Each participant performed three consecutive maximal voluntary contractions for both concentric and eccentric muscle actions at each angular velocity, with a 90-s rest between tests at different velocities to minimize fatigue. All tests were supervised by a physiotherapist, who ensured proper technique, alignment, and maximal effort, and instructed participants to avoid compensatory movements (such as excessive trunk rotation or shoulder elevation) that could compromise measurement validity. The order of testing (angular velocity and internal vs. external rotation) was randomized to avoid order effects. For each condition, the mean of the peak torque from the three maximal repetitions was calculated and used for statistical analysis.

#### 2.4.2. Serve Speed and Kinematic Variables

Following a standardized 15-min dynamic warm-up, participants performed the serving protocol. The warm-up aimed to optimize neuromuscular readiness and reduce the risk of acute shoulder injury and consisted of 5 min of light aerobic activity (jogging) with dynamic lower-body movements (calf raises, hip hinges, lunges, squats, multidirectional hopping), followed by 10 min of upper-body activation targeting the glenohumeral and scapular stabilizers. Resistance band exercises included external and internal rotations at 90° shoulder abductions (3 × 10 repetitions), scapular retractions (3 × 10 reps), and diagonal patterns (3 × 10 reps per arm). To ensure familiarity with the procedure and minimize learning effects, each participant completed five submaximal serves toward the designated target zone prior to testing. The experimental protocol consisted of two series of 20 maximal-effort flat serves, delivered toward the deuce side of the court (from a right-handed perspective) in accordance with ITF rules. Each series was separated by a 3-min passive rest and subdivided into four sets of five serves with 30-s rests to reduce fatigue accumulation. Verbal cues emphasized the generation of maximal power and directional control.

Serve speed was measured using a calibrated radar gun (Stalker Pro II, Applied Concepts Inc., Richardson, TX, USA) positioned behind the server on the opposite baseline at a height of 1.5 m, directly aligned with the ball trajectory. The radar operated at a sampling frequency of 33 Hz and recorded the ball’s peak velocity during the initial free-flight phase immediately following racket–ball contact. Radar and motion-capture data were collected concurrently and paired by trial ID, without electronic synchronization. The geometric limitations of radar-based measurements have been previously described, indicating that line-of-sight misalignment can underestimate true ball velocity by ~3–4% at the serve impact point and up to 14% on court bounce when alignment is suboptimal [36]. In the present setup, the radar was aligned with the central flight path of the ball to minimize angular error. The target zone was a 1 × 2 m rectangle located 1 m laterally from the center service line and extending 2 m toward the net from the service line, marked with white floor tape. This zone size was selected to represent a realistic tactical serve placement and is consistent with previous analyses of serve accuracy in competitive players [21]. From the 40 recorded serves, the first 10 successful serves (landing within the target zone) and the first 10 unsuccessful serves (landing outside or into the net) were analyzed to enable direct comparison between effective and ineffective executions. Here, “first 10” denotes the earliest 10 instances per outcome category, not the first 10 serves in chronological order. Because successful and unsuccessful serves occurred in varying sequences, the subsets did not necessarily follow the same temporal order. This approach ensured a balanced representation of both serve outcomes while maintaining biomechanical consistency and minimizing potential order, learning, or fatigue effects across 40 trials. Preliminary inspection of trial-level serve speeds indicated no systematic temporal decline across attempts, suggesting negligible order effects. This within-player comparison aligns with previous biomechanical research on tennis serve performance [15,21,24].

Kinematic data were collected using a calibrated six-camera three-dimensional motion capture system (Version 2025.2; Qualisys AB, Gothenburg, Sweden) operating at a sampling rate of 200 Hz (5 ms frame interval). Calibration included L-frame spatial calibration and dynamic wand calibration prior to each session, yielding a mean residual spatial error of less than ±1 mm. The global coordinate system was defined by the L-frame; for reporting vertical variables, heights were referenced to the lead foot’s fifth metatarsal marker at stance (z = 0). Reflective markers were placed at key equipment landmarks: three small circular patches (8 mm diameter) cut from lightweight reflective tape were symmetrically attached to the tennis ball to ensure constant visibility during ball rotation while adding negligible mass (<0.5 g; <0.4% of ball mass). Comparable marker configurations have been used previously without observable influence on ball trajectory or rotation [21]. Four markers were positioned on the racket (tip of the head, each lateral side of the frame, and base of the handle). These facilitated the computation of zenith height, impact height, and the zenith–impact differential. The moment of ball–racket contact was defined as the first captured frame showing intersection or compression between the racket and ball marker clusters. Given a 200 Hz sampling rate (5 ms per frame), this timing corresponds to a temporal uncertainty of approximately ±1–2 frames (±5–10 ms), which translates to a vertical uncertainty of roughly ±2 mm. No marker occlusion occurred during data collection. In the rare case of transient marker loss (less than 3 frames), cubic-spline interpolation in Qualisys Track Manager was applied. The origin of the coordinate system was defined by the marker placed on the dorsal aspect of the fifth metatarsal of the lead foot. All testing was supervised by a three-member research team: one operated the radar gun and provided verbal serve speed feedback, one logged serve outcomes and marker visibility, and one fed ball and monitored rule compliance.

### 2.5. Data Collection

Primary outcome variables included peak torque values (Nm) for concentric and eccentric internal and external shoulder rotations at angular velocities of 210°/s and 300°/s, measured using the isokinetic dynamometer. For each condition, the mean peak torque across three maximal trials was calculated and used in subsequent analyses. Serve speed (km/h) was calculated as the mean of the first 10 successful serves (landing within the target zone) and the mean of the first 10 unsuccessful serves (landing outside the target zone or into the net) from each participant. This selection was made to standardize comparisons while minimizing potential effects of fatigue across the 40 recorded serves. Kinematic variables were extracted from three-dimensional motion capture data processed in Qualisys Track Manager (Version 2025.2; Qualisys AB, Gothenburg, Sweden). Zenith height was defined as the maximum vertical position (m) of the ball marker during the toss phase, and impact height as the vertical position (m) of the ball marker at the frame corresponding to the racket-ball contact. For sensitivity analysis, a normalized impact height was also calculated as the impact height divided by the player’s body height. The zenith–impact differential (m) was computed as the vertical distance between zenith height and impact height. The global coordinate system was defined by L-frame calibration; for reporting vertical variables, heights were referenced to the lead foot’s fifth metatarsal marker at stance (z = 0). The motion capture system had a spatial accuracy of ±1 mm, and all vertical measurements were referenced to the laboratory coordinate system defined during the spatial calibration process. Toss-execution variability (moderators) was defined as the within-player standard deviation (SD) of impact height and the SD of the zenith–impact differential, computed separately across the 10 successful serves and across the 10 unsuccessful serves. The radar and motion-capture systems were operated independently, with serve speed corresponding to the same serve trial identified in the kinematic sequence. All kinematic data were processed by a single operator using a standardized analysis pipeline in Qualisys Track Manager, with automatic extraction of marker trajectories.

### 2.6. Statistical Analysis

All statistical analyses were performed using IBM SPSS Statistics for Windows (version 25.0; IBM Corp., Armonk, NY, USA) and Microsoft Excel 2019 (version 2312; Microsoft Corp., Redmond, WA, USA). Prior to inferential analyses, data distributions were examined for normality using the Shapiro–Wilk test. Given the limited sample size (*n* = 13) and observed deviations from normality in some variables, bivariate associations were consistently evaluated using Spearman’s rank-order correlation (ρ), with corresponding *p*-values reported. Correlation analyses assessed the relationships between serve speed (successful and unsuccessful trials), isokinetic shoulder strength (mean peak torque at 210°/s and 300°/s, for both concentric and eccentric actions), and kinematic variables (zenith height, impact height, and the zenith–impact differential). Scatter plots with regression lines and 95% confidence bands were generated to aid interpretation.

To address the main hypotheses, three exploratory mediation models were tested to evaluate whether kinematic variables mediated the relationship between shoulder internal rotation strength and serve speed: (1) zenith height as mediator, (2) impact height as mediator, and (3) zenith–impact differential as mediator. Analyses were conducted using the PROCESS macro for SPSS (Version 4.3; Model 4; Andrew F. Hayes, Calgary, AB, Canada) with bias-corrected and accelerated bootstrapping (5000 samples) to generate 95% confidence intervals for indirect effects [37]. In additional analysis, players’ height was included as a covariate to control for anthropometric differences. All continuous variables were standardized prior to mediation and moderation analyses to facilitate interpretation and simple-slopes probing. Moderation analyses were conducted to examine whether toss consistency (within player standard deviation of zenith height, impact height, or zenith–impact differential) moderated the relationship between shoulder-rotator strength and serve speed. The PROCESS macro (Version 4.3; Model 1; Andrew F. Hayes, Calgary, AB, Canada) was used to test interaction terms (X × Z, where X represents shoulder strength and Z represents toss variability). Significant interactions were probed with simple slopes analyses at ±1 SD of the moderator.

Primary predictors (isokinetic strength measures) and moderators (toss-consistency metrics; within-session SDs) were defined at the participant level. To align the level of measurement and avoid pseudo-replication, serve-speed outcomes were summarized per participant (means of the first ten successful and first ten unsuccessful serves), and associations were estimated at the subject level. Since the analyses addressed a small, theory-driven set of a priori hypotheses (correlation between shoulder-rotator strength and serve speed; mediation by zenith, impact, or differential; moderation by toss variability), no formal multiplicity correction was applied. Given the limited sample and high detectable-effect thresholds, conventional family-wise or FDR adjustments would markedly reduce power. Therefore, results are presented with effect sizes and 95% confidence intervals and interpreted as exploratory.

Due to the limited sample size (*n* = 13), we conducted a sensitivity power analysis using G*Power (version 3.1.9.7; Heinrich Heine University Düsseldorf, Düsseldorf, Germany) for all planned tests. In the case of (Exact) bi-variate correlations, with α = 0.05 and β = 0.20 (80% power) for a two-tailed test, the minimum detectable effect size was ρ ≥ 0.69. Thus, only correlations at or above this magnitude could be reliably detected in the present sample. For mediation analyses (*n* = 13), only very large indirect effects (i.e., both paths a and b large) could reach 80% power, whereas smaller mediated effects would likely remain undetected. For moderation analyses (linear models including X, Z, and X × Z interaction), assuming α = 0.05, power = 0.80, and df_2_ = 9, the detectable incremental variance explained by the interaction term corresponds to a large effect (ΔR^2^ ≈ 0.30–0.40; Cohen’s f^2^ ≈ 0.45–0.65). Therefore, the present study should be regarded as exploratory, and non-significant results, particularly in the mediation and moderation models, should be interpreted with caution due to limited statistical power.

## 3. Results

Descriptive statistics for all key variables are summarized in Table 1. On average, successful serves reached 162.65 ± 8.03 km/h, while unsuccessful serves reached 161.16 ± 7.70 km/h. Mean zenith height was 3.404 ± 0.243 m for successful and 3.397 ± 0.253 m for unsuccessful serves (Δ ≈ 0.007 m). Mean impact height was 2.680 ± 0.128 m (successful) and 2.677 ± 0.116 m (unsuccessful), and the zenith–impact differential averaged 0.724 ± 0.189 m and 0.720 ± 0.190 m, respectively. These values indicate only minimal between-condition differences, suggesting that the binary outcome (successful vs. unsuccessful) reflects a subtle performance distinction rather than a substantial biomechanical contrast. Given the limited sample size (*n* = 13) and corresponding a priori sensitivity (80% power to detect ρ ≥ 0.69), smaller associations may have remained undetected; therefore, subsequent analyses are interpreted as exploratory.

Robustness check. To verify that the trial-selection approach did not bias the outcomes within individual players, we compared per-player averages derived from the first 10 trials against those based on all available repetitions. Across players, the mean absolute within-subject difference in serve speed between the two calculation methods was 0.3 ± 0.6 km/h for successful and 0.4 ± 0.7 km/h for unsuccessful serves (maximum deviation < 1.2 km/h). These trivial differences confirm that the selection of the first 10 valid trials did not meaningfully alter individual serve-speed estimates or between-condition contrasts. Stability of variability estimates. To verify that the toss-variability metrics (SD of zenith height, impact height, and zenith–impact differential) were stable despite being derived from 10 trials, a leave-one-out (jackknife) sensitivity analysis was performed. The mean change in SD after removing one trial ranged from 3 to 6 mm (≈ 5–8% of baseline SD) across variables and conditions, indicating that the variability measures were robust against single-trial omission.

### 3.1. Correlation Analysis Between Kinematics Variables and Isokinetic Strength or Serve Speed

Since several variables deviated from normality (Shapiro–Wilk), Spearman’s ρ was used for bivariate associations. For successful serves, impact height showed a strong positive association with serve speed (ρ = 0.746, 95% CI [0.31, 0.93], *p* = 0.003, *n* = 13; Table 2). When impact height was normalized to player height (impact height/height), this association remained significant (ρ = 0.627, 95% CI [0.06, 0.89], *p* = 0.022, *n* = 13), although the confidence interval was wide and the study was powered to reliably detect only |ρ| ≥ 0.69; therefore, this result should be interpreted cautiously. However, this indicates that the relationship is not solely driven by anthropometric differences. In contrast, zenith height showed only a weak, non-significant association with serve speed (ρ = 0.289, 95% CI [−0.31, 0.73], *p* = 0.338), and the zenith–impact differential was similarly non-significant (ρ = −0.217, 95% CI [−0.69, 0.41], *p* = 0.476). Correlations between isokinetic shoulder strength (internal and external rotation at 210°/s and 300°/s, concentric and eccentric modes) and serve speed were also non-significant (all *p* ≥ 0.094). Interpretations are exploratory, given the study’s sensitivity (see Table 2 notes).

A similar pattern emerged for kinematic variables in unsuccessful serves. Impact height correlated strongly and significantly with serve speed (ρ = 0.776, 95% CI [0.36, 0.94], *p* = 0.002; Table 3). When impact height was normalized to body height (impact height divided by height), the association weakened to a moderate, non-significant level (ρ = 0.518, 95% CI [−0.10, 0.85], *p* = 0.070, *n* = 13), suggesting that part of the raw relationship is attributable to body height. In contrast, zenith height showed only a weak, non-significant correlation with serve speed (ρ = 0.269, 95% CI [−0.33, 0.72], *p* = 0.375), and the zenith–impact differential was similarly negligible (ρ = −0.030, 95% CI [−0.57, 0.53], *p* = 0.921). Two shoulder strength variables were positively associated with serve speed: internal rotation eccentric 210°/s (ρ = 0.615, 95% CI [0.03, 0.88], *p* = 0.025) and external rotation eccentric 300°/s (ρ = 0.601, 95% CI [0.00, 0.87], *p* = 0.030). All other strength–speed correlations were non-significant (*p* ≥ 0.094). Interpretations are exploratory, given the study’s sensitivity (see Table 3 notes).

Across both serve outcome conditions, none of the isokinetic strength variables correlated significantly with any kinematic parameters (Table 2 and Table 3). Given the study’s sensitivity (80% power to detect |ρ| ≥ 0.69; see table notes), only large effects could be reliably observed; therefore, non-significant correlations are inconclusive. Overall, impact height consistently relates to serve speed, whereas shoulder-rotator strength shows limited direct associations with kinematic variables. This pattern motivated the subsequent mediation and moderation analyses to test whether kinematic variables, particularly impact height and their variability, act as mediators or moderators linking shoulder-rotator strength to serve speed.

### 3.2. Graphical Representation of the Results

Scatterplots (Figure 2 and Figure 3) illustrate the association between serve speed and impact height. In absolute terms (Figure 2), there is a positive linear trend for both successful (Figure 2a) and unsuccessful (Figure 2b) serves. The fitted OLS lines have similar positive slopes with comparable fit (R^2^ ≈ 0.62 and 0.60, respectively). These regressions are descriptive only; statistical inference is based on Spearman’s ρ. This indicates that higher impact points are, on average, associated with faster serves across outcomes. When impact height was normalized to body height (Figure 3), the positive trend remained; suggesting that the relationship is not solely driven by differences in body height. The correlations reported in Section 3.1 confirmed significant positive associations for absolute impact height in both conditions and for relative impact height only in successful serves. Taken together, the plots and correlations suggest that attaining a higher contact point is linked with greater serve velocity, with weaker evidence for the relative measure in unsuccessful trials. Given the small sample size (*n* = 13), the confidence bands are relatively wide, and the relationships should be interpreted cautiously as associative, rather than causal.

Figure 4 presents the association between eccentric internal shoulder-rotator strength and impact height during successful serve attempts, evaluated at two angular velocities: 210°/s (Figure 4a) and 300°/s (Figure 4b). In both panels, the fitted OLS lines show a positive trend; these fits are descriptive, and statistical inference is based on Spearman’s ρ, which was non-significant (*p* > 0.05). Despite the positive slopes, the relatively low R^2^ values (≈0.20 and 0.14) indicate weak linear associations. Accordingly, while eccentric shoulder strength may relate to slightly higher impact heights, it alone explains only a small portion of the variance.

### 3.3. Mediation Analyses

To examine whether kinematic parameters transmit the association between strength and performance, we specified three simple mediation models with internal shoulder rotation strength (eccentric, 210°/s) as the predictor (X) and serve speed as the outcome (Y). Each model entered one kinematic mediator (M) in turn: impact height, zenith height, or the zenith–impact differential. All continuous variables were standardized (coefficients reported as *β*). Indirect effects (a × b) were estimated using bias-corrected and accelerated bootstrapping (5000 iterations; 95% CIs). In the a-path notation, “a” is the effect of X on M, “b” is the effect of M on Y controlling for X, and “c” is the direct effect of X on Y controlling for M (Figure 5).

Successful serves. In the model with impact height, the a-path was positive but non-significant (*β* = 0.450, *p* = 0.123, 95% CI [−0.13, 0.81]), whereas the b-path was positive and significant (*β* = 0.699, *p* = 0.008, 95% CI [0.23, 0.90]). Thus, players who contacted the ball higher tended to achieve higher serve speeds even after accounting for shoulder strength, but shoulder strength itself did not reliably predict impact height. The indirect effect was a×b = 0.315 with 95% CI [−0.083, 0.737] (bootstrap two-tailed *p* = 0.124; 5000 resamples), and the direct effect *c′* was small and non-significant (*β* = 0.193, *p* = 0.380, 95% CI [−0.28, 0.63]). Explained variance was moderate (R^2^ = 0.49 for the mediator model and R^2^ = 0.61 for the outcome model). For the other mediators, indirect effects were likewise non-significant: zenith height (a = −0.015; b = 0.397; a × b = −0.006; 95% CI [−0.396, 0.334]) and zenith–impact differential (a = −0.324; b = 0.149; a × b = −0.048; 95% CI [−0.234, 0.365]).

Unsuccessful serves. With impact height as mediator, coefficients were a = 0.331 (95% CI [−0.24, 0.74]) and b = 0.654 (95% CI [0.15, 0.88]), yielding an indirect effect a × b = 0.217 with 95% CI [−0.173, 0.681] (non-significant); the direct effect *c′* was positive but non-significant (*β* = 0.360, *p* = 0.112, 95% CI [−0.21, 0.74]). Mediation via zenith height and zenith–impact differential was also non-significant.

Across models, none of the indirect-effect confidence intervals excluded zero, indicating that in this sample the tested kinematic variables did not reliably mediate the strength–speed relationship. Practically, although a higher impact height is directly associated with greater serve speed (significant b-path), eccentric internal shoulder-rotator strength did not consistently lead to a higher impact height (non-significant a-path). When players’ height was added as a covariate, the indirect effect of impact height remained non-significant and further attenuated (successful serves: *β*_indirect ≈ 0.09; unsuccessful serves: *β*_indirect ≈ −0.03). Thus, no evidence of mediation was found even after controlling for players’ height.

### 3.4. Moderation Analyses

We tested whether toss consistency moderates the association between eccentric internal shoulder-rotator strength (eccentric, 210°/s) and serve speed. Toss consistency was operationalized as the standard deviation (SD) of impact height and the zenith–impact differential. All predictors were standardized prior to model estimation; interaction terms were probed with simple slopes at ±1 SD of the moderator.

Successful serves. The interaction between strength and impact-height variability was significant (*β*_int = −0.417, SE = 0.18, *t* = −2.49, *p* = 0.043, 95% CI [−0.81, −0.02], *R^2^* = 0.551). Simple slopes showed a strong, positive strength–speed association under low variability (−1 SD; *β* = 1.03, *p* = 0.015, 95% CI [0.25, 1.81]), but a weak, non-significant association under high variability (+1 SD; *β* = 0.19, *p* = 0.488, 95% CI [−0.38, 0.76]). Likewise, the interaction with zenith–impact differential variability was significant (*β*_int = −0.493, SE = 0.21, *t* = −2.33, *p* = 0.048, 95% CI [−0.98, −0.01], *R^2^* = 0.533). Simple slopes again indicated a robust positive association at low variability (−1 SD; *β* = 1.22, *p* = 0.012, 95% CI [0.31, 2.12]) that attenuated at high variability (+1 SD; *β* = 0.24, *p* = 0.384, 95% CI [−0.39, 0.87]). These patterns (Figure 6) indicate that greater toss variability diminishes the extent to which shoulder-rotator strength translates into ball speed during successful serves. A summary of the standardized coefficients and model statistics for both moderation analyses is presented in Table 4. Unsuccessful serves. No interaction terms reached significance (all *p* > 0.25), suggesting that, for unsuccessful trials, toss variability did not materially alter the relation between strength and serve speed.

The negative interaction coefficients (*β*_int < 0) consistently show that higher toss variability weakens the otherwise positive link between internal eccentric rotation strength and serve speed. Practically, consistent toss execution appears necessary for players to convert shoulder strength into faster serves. Given the small sample size (*n* = 13) and limited power for the interaction effects (detectable ΔR^2^ ≈ 0.30–0.40), these moderation findings are exploratory.

## 4. Discussion

The present study examined how toss-phase kinematics interact with shoulder rotation strength to influence serve performance in junior tennis players and extended prior work [29]. Consistent with kinetic-chain models emphasizing sequential force transfer and timing [9,10], and prior evidence linking toss quality to serve outcomes [15,16,17,25], we expected the impact height to contribute to serve-speed production. Our findings partially supported the association between impact height and serve speed in both successful and unsuccessful serves. This association remained significant after impact height was normalized to the player’s height for successful serves, indicating that the link is not purely anthropometric. In contrast, zenith height and the zenith–impact differential were unrelated to serve speed, in line with previous findings in elite juniors [38]. These results refine earlier work by emphasizing that impact height, but not all toss parameters, acts as a meaningful kinematic indicator of serve efficiency in junior players.

### 4.1. Association Between Kinematic Toss Variables and Shoulder Rotation Strength or Serve Speed

Correlation analyses provided further insights into the interplay between shoulder strength, kinematic parameters, and serve speed. In this sample, most strength–speed correlations were not statistically significant; only two eccentric measures (Internal rotation at 210°/s and External rotation at 300°/s). Prior work has documented significantly greater dominant arm internal rotation torque compared to the non-dominant arm in elite junior tennis players [27].

Consistent with prior work on toss quality/consistency and contact-point characteristics [13,15,16,17,25,32], impact height showed a positive association with serve speed for successful and unsuccessful serves in our sample of junior tennis players. Because taller players tend to reach higher contact points and serve faster [9,30], we checked that when impact height was normalized to players’ height, the association remained significant for successful serves, indicating a technical component beyond anthropometry. By contrast, zenith height and the zenith–impact differential were not significantly related to serve speed, and none of the kinematic variables showed a reliable association with accuracy under the present binary definition (successful vs. unsuccessful). Taken together, these results support the view that the location of ball–racket contact is determinant of racket-head velocity, whereas toss geometry per se has limited predictive value unless integrated into a well-timed kinetic sequence [3,23,24]. Finally, no isokinetic strength variable correlated with the kinematic metrics (impact height, zenith height, and zenith–impact differential), suggesting that the influence of shoulder-rotator strength on performance emerges through coordination and execution consistency rather than through simple mapping onto contact geometry.

### 4.2. The Mediating Role of Impact Height

We originally posited impact height as a plausible biomechanical mediator transmitting the effect of internal shoulder-rotator strength to serve speed, consistent with kinetic-chain accounts of proximal-to-distal energy flow and with evidence that contact-point characteristics are closely tied to serve outcomes [3,13,21,23,24,32]. In junior cohorts, higher impact height has been associated with faster serve speed and related to strength capacities, particularly lower-limb power, supporting the rationale for a strength–impact height–serve speed pathway [39]. In our models, the b-path from impact height to speed (controlling strength) was significant, but the a-path from strength to impact height was not, yielding a non-significant indirect effect (a × b). Thus, while a higher contact point was directly associated with faster serves (and remained so when impact height was normalized to players’ height for successful serves), we did not find statistically significant evidence that greater shoulder strength reliably produces a higher impact point in this cohort. Mechanistically, this is coherent with the view that a higher contact point reduces the need for compensatory trunk/arm adjustments and supports coordinated timing within the kinetic chain [15,21,24], yet achieving that favorable contact still depends on how the motion is organized rather than on isolated torque alone [3,23,24].

Importantly, the present findings suggest a shift in emphasis for practice: rather than focusing on toss geometry in isolation, technical work could prioritize consistently attaining an efficient, elevated impact point, the proximal determinant of racket-head speed, while strength and power training are integrated to support that contact under repeatable timing [13,15,24,32]. In our data, this integration is echoed by the moderation results. When toss-execution consistency is high (lower SD of impact height or the zenith–impact differential), the positive strength–speed slope is expressed. Together, these patterns refine the mediation hypothesis that impact height matters for serve speed, but in this sample, it did not mediate the strength–speed link. Rather, execution consistency appears to be the conditional factor that allows shoulder-rotator strength to translate into faster serves.

Although impact height can be considered a proximal determinant of serve velocity, this relationship operates within a complex coordination system. The present data do not capture additional determinants such as lateral toss displacement, racket-face orientation, or the timing of trunk and pelvic rotations, all of which may critically influence how kinetic-chain energy is transferred to the ball [8,12,15,22]. Serve velocity emerges from the temporal sequencing and intersegmental coordination of the lower limbs, trunk, and upper extremity rather than from any single mechanical variable [13,24,32,38]. Thus, the observed effects of strength and impact height should be interpreted as components of a broader coordination process rather than as isolated causal factors. In this sense, strength provides mechanical potential, while coordination dictates its effective use [12,15,24].

However, the absence of a statistically significant indirect effect should not be interpreted as evidence that no mediation exists. Given the small sample size and limited statistical power (detectable ρ ≥ 0.69), smaller indirect effects may have remained undetected. Therefore, the present findings indicate a lack of evidence for mediation rather than evidence of its absence, underscoring the exploratory character of the study. Importantly, the mediation effect of impact height did not reach significance even when players’ height was statistically controlled. This suggests that the non-significant mediation was not an artifact of anthropometric differences but rather reflects the limited contribution of impact height as an intermediary between shoulder strength and serve speed in this junior cohort.

### 4.3. Toss Execution Consistency as a Moderator

Moderation analyses showed that toss variability (operationalized as the SD of impact height and of the zenith–impact differential across trials) significantly moderated the strength–speed association for successful serves. This moderating effect was observed only for successful serves. The absence of a similar pattern in unsuccessful serves may reflect self-selection of more optimal contact conditions during effective trials or limited statistical power to detect smaller effects. Specifically, simple-slopes tests indicated a positive strength–speed slope under low variability but a flat, non-significant slope under high variability. In practical terms, greater toss consistency amplifies the translation of shoulder-rotator strength into ball speed, whereas erratic kinematic execution dampens or nullifies this translation. This pattern aligns with work linking toss consistency and contact-point control to serve outcomes and with kinetic-chain accounts emphasizing that high racket-head speed emerges from repeatable timing and coordinated proximal-to-distal sequencing, rather than from isolated torque alone [8,22]. These findings also help reconcile mixed strength–speed correlations by indicating that strength is necessary but not sufficient: without stable toss execution, its effect on speed can be attenuated [28]. Collectively, the moderation results highlight toss-execution consistency as a conditional amplifier that enables available shoulder-rotator strength to manifest as higher serve speed. Given the small sample size, these findings should be interpreted as exploratory rather than confirmatory. Nevertheless, the observed interaction suggests that toss-execution consistency may represent a practical condition under which shoulder-rotator strength translates into higher serve speed. In the context of junior tennis players, who often display greater motor control variability, these results underscore the importance of stable toss mechanics in achieving strength-based performance advantages.

### 4.4. Successful Versus Unsuccessful Serves

Across conditions, serve speed and impact height were nearly identical (successful vs. unsuccessful: impact height ≈ 2.680 m vs. 2.677 m), indicating that neither serve speed nor vertical contact point alone discriminated serve outcome in this cohort. The only kinematic difference was a slightly higher toss zenith in successful serves (≈ +0.7 cm on average), which yielded a trivially larger zenith–impact differential. Although some frameworks would predict that a smaller zenith–impact gap reflects more efficient timing [15,25], the magnitude of the observed differences here was small, suggesting that vertical geometry by itself is unlikely to determine success at this developmental stage. Rather, success likely depends on how contact is organized in space and time (racket-face orientation, lateral toss placement, and location-specific kinematics), factors not captured by our vertical-only metrics [21,32]. This interpretation is consistent with evidence that toss consistency, and contact-point control relate to serve outcomes [15,21,25], and that spatial characteristics (aiming/location patterns) contribute to success beyond ball speed per se [32]. Methodologically, our binary accuracy definition (first 10 successful vs. first 10 unsuccessful serves) may also have limited sensitivity relative to continuous placement error, and the predefined target zone could have attenuated variance. Together with our moderation results, where toss-execution consistency conditioned the translation of strength into speed for successful serves, these findings suggest that precision and repeatability in temporal–spatial execution (not simply higher speed or higher contact) are pivotal for converting physical capacity into successful serve outcomes in junior players [3,21,24,32].

### 4.5. Practical Applications

Our results reinforce a kinetic-chain view of the serve, in which racket-head speed emerges from repeatable, well-timed proximal-to-distal sequencing rather than from isolated joint torque [3,23,24]. The contact point (impact height) appears to be a proximal indicator of serve speed, whereas toss geometry, per se, has limited predictive value unless it supports achieving a stable, elevated impact height [13,15,21,25,32]. Finally, the observed moderation by toss-execution consistency indicates that strength is necessary, but its effective translation into serve speed requires sufficient stability of the technical execution. The translation of available shoulder-rotator strength into ball speed is amplified when variability in impact height or the zenith–impact differential is low [4,11,22]. Therefore, coaching practices could aim to pair shoulder rotation strength development with contact-point stabilization and toss-consistency drills, aligning technical work with the principles of the kinetic-chain. For monitoring, practitioners may consider tracking the impact height and its variability as indicators of technical consistency and readiness to express strength in serve performance.

### 4.6. Limitations of the Study

This study has several limitations that should be considered when interpreting the findings. First, the sample was small and homogeneous (*n* = 13 nationally ranked junior males), which reduces power. No a priori power calculation was performed due to the limited availability of elite junior players; instead, a post-hoc sensitivity analysis was conducted. This analysis indicated that only relatively large correlations (ρ ≥ 0.69) could be detected with 80% power. For mediation, only large indirect effects would be detectable according to simulation-based estimates. For moderation, only large interaction effects (ΔR^2^ ≈ 0.30–0.40) would reach adequate power. Given the small sample size and corresponding wide CIs, these mediation findings, particularly from the mediation and moderation analyses, should be interpreted cautiously and viewed as exploratory and hypothesis-generating for larger studies. Additionally, bootstrapped mediation estimates in small samples may yield unstable confidence intervals, and therefore, their precision should be interpreted cautiously. Future staged validation should include multi-site cohorts and female athletes to confirm the robustness and generalizability of these findings. Second, the design was cross-sectional. As such, the observed relationships cannot establish causality (whether increases in strength lead to higher impact points or faster serves). Third, measurement choices prioritize control and reproducibility over ecological validity. Although isokinetic testing in a supine position with the shoulder at 90° abduction (ROM set to 90% ER and 65% IR) provided standardized and safe assessment of shoulder torque, this setup does not fully replicate the ballistic stretch–shortening cycle, elastic energy storage, or extreme external rotation angles characteristic of a live serve. Furthermore, using the mean of three trials may slightly under-represent a player’s absolute peak performance. Conversely, our kinematic model focused on vertical metrics (zenith height, impact height, zenith–impact differential). We did not quantify lateral/anterior–posterior toss displacement, time-to-impact, or temporal coupling with segmental rotations, all of which likely contribute to contact quality. Future analyses should therefore integrate additional kinematic dimensions, particularly lateral toss displacement, racket-face orientation at impact, and temporal proximity to contact, to capture the multidimensional nature of toss control. Incorporating these parameters through advanced 3D motion capture or validated markerless tracking could clarify how spatial and temporal variability jointly influence serve success. Motion capture at 200 Hz is adequate for toss but is near the lower bound for pinpointing impact events; small timing errors could add noise to impact height estimates. Future studies could address this limitation by increasing capture frequency (e.g., ≥500–1000 Hz) or by synchronizing motion capture with an external impact-detection system, such as a racket-mounted accelerometer, acoustic trigger, or optical contact sensor. These solutions would improve temporal resolution and reduce uncertainty in identifying the exact moment of impact. In addition to optical precision, instrumentation alignment also affects measurement accuracy. Although the radar was aligned with the ball’s trajectory, cosine error cannot be fully excluded. Finally, the radar and motion-capture systems were operated independently without electronic synchronization. Although this may introduce minimal temporal uncertainty in aligning serve-speed and kinematic data, the potential offset (within tens of milliseconds) was considered negligible relative to the 5 ms frame interval of motion capture and the measurement precision of both systems. Fourth, operationalization of performance may have constrained sensitivity. Accuracy was treated as binary (successful vs. unsuccessful) within a predefined target zone. Continuous placement error might detect subtler effects. We analyzed the first 10 successful and first 10 unsuccessful serves from 40 attempts. While this standardized sample size across players, it may introduce selection/order effects and reduce the amount of trial-level data available for estimating within-player variability. Fifth, confounding and collinearity remain concerns. Body height correlates with impact height and serve speed, which can inflate bivariate associations. We addressed this by reporting height-normalized impact height, but residual confounding by other dimensions (e.g., limb segment lengths) remains possible. While subject-level aggregation preserves construct alignment (subject-level strength and toss-consistency) and avoids pseudo-replication, it sacrifices trial-level information. Future studies with larger, multi-site cohorts should employ trial-level mixed-effects models (random intercepts for players, trial-level kinematics, and placement) to jointly estimate within and between-subject effects, thereby mitigating aggregation bias.

## 5. Conclusions

Impact height showed a consistent positive association with serve speed. However, mediation of the relationship between shoulder-rotator strength and serve speed via impact height was not statistically supported, whereas toss-execution consistency significantly moderated this link (for successful serves). Practically, training approaches could focus on stabilizing and elevating the impact point and reducing toss variability to enhance the conversion of shoulder-rotator strength into ball speed. However, these strategies should be implemented cautiously, given the exploratory and cross-sectional nature of the data. Overall, the findings should be regarded as preliminary and require replication in larger and more heterogeneous samples. Future work could extend this approach by incorporating three-dimensional toss parameters, additional kinematic variables, and comparative analyses across age groups and female players to validate and generalize the present framework.

## Figures and Tables

**Figure 1 jfmk-10-00438-f001:**
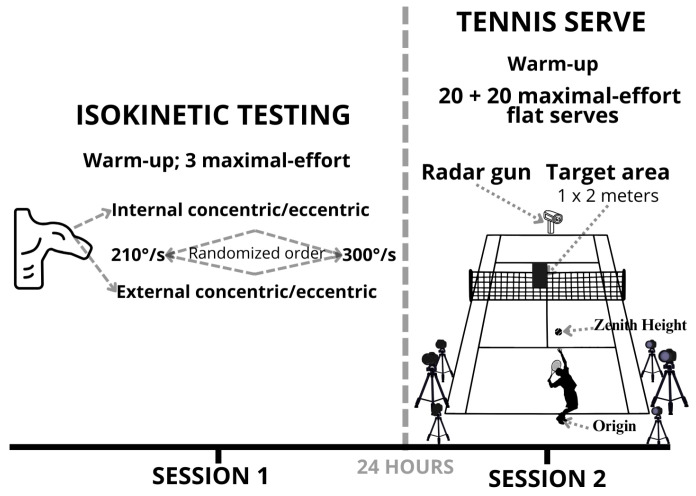
Overview of the testing protocol.

**Figure 2 jfmk-10-00438-f002:**
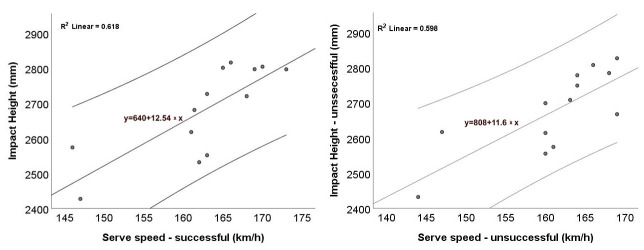
Relationship between serve speed (km/h) and impact height (mm): (**a**) successful; (**b**) unsuccessful serves. Each dot represents a participant’s mean from the first 10 successful or first 10 unsuccessful serves recorded in that category (i.e., the earliest 10 instances of each outcome, not necessarily the first 10 serves overall). The solid line shows the OLS linear fit + 95% CI. Note that OLS lines are descriptive; inference is based on Spearman’s ρ.

**Figure 3 jfmk-10-00438-f003:**
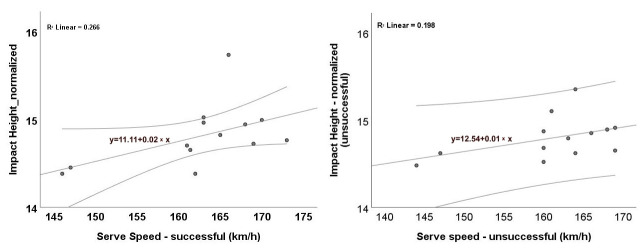
Relationship between serve speed (km/h) and impact height (normalized): (**a**) successful; (**b**) unsuccessful serves. Each dot represents a participant’s mean from the first 10 successful or first 10 unsuccessful serves recorded in that category). The solid line shows the OLS linear fit + 95% CI. Note that OLS lines are descriptive; inference is based on Spearman’s ρ.

**Figure 4 jfmk-10-00438-f004:**
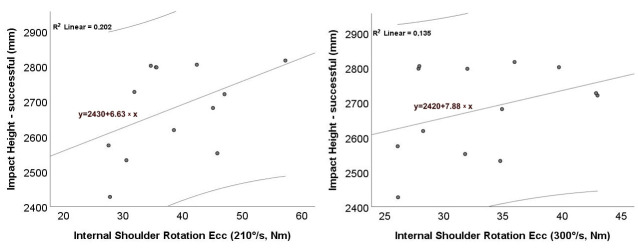
Relationship between eccentric internal shoulder-rotator strength (peak torque, Nm) and impact height (mm) during successful serves: (**a**) 210°/s; (**b**) 300°/s. Each dot represents a participant’s mean from the first 10 successful serves (the earliest 10 instances meeting the success criterion). The solid line shows the OLS linear fit + 95% CI. Note that OLS lines are descriptive; inference is based on Spearman’s ρ (n = 13). Ecc—eccentric.

**Figure 5 jfmk-10-00438-f005:**
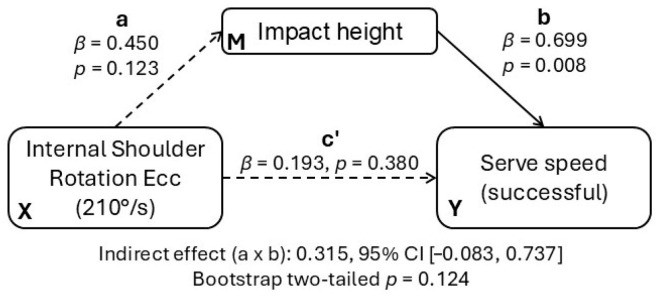
Mediation model for successful serves (standardized coefficients). X = eccentric internal shoulder-rotator strength at 210°/s; M = impact height; Y = serve speed. Path “a” denotes the effect of X on M; path “b” denotes the effect of M on Y, controlling for X; path “c′” denotes the direct effect of X on Y, controlling for M. Numbers shown next to paths are standardized *β* and two-tailed *p* values. The indirect effect (a × b) was estimated using a bias-corrected and accelerated bootstrapping (5000 resamples) and is reported with 95% CI and two-tailed *p*; *n* = 13. Ecc—eccentric. (Dashed arrows indicate non-significant paths).

**Figure 6 jfmk-10-00438-f006:**
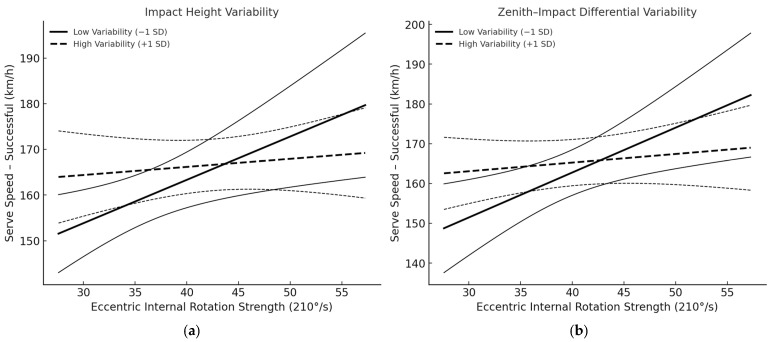
Simple slopes of the relationship between eccentric internal shoulder-rotator strength (210°/s) and serve speed at low (−1 SD) and high (+1 SD) levels of toss variability for successful serves: (**a**) Impact-height variability; (**b**) Zenith–impact differential variability. Solid and dashed lines represent regression slopes at −1 SD and +1 SD of the moderator, respectively. Thin flanking lines denote the 95% confidence bands. Lines show predicted values from linear models with standardized predictors including the X × Z interaction.

**Table 1 jfmk-10-00438-t001:** The performance of kinematic variables, isokinetic net moment, and serve speed.

Variables	Mean	SD	95% CI	SW
Zenith height—successful serve (m)	3.404	0.24	3.257–3.550	0.84
Zenith height—unsuccessful serve (m)	3.397	0.25	3.244–3.549	0.77
Impact height—successful serve (m)	2.680	0.13	2.602–2.757	0.10
Impact height—unsuccessful serve (m)	2.677	0.12	2.607–2.747	0.58
Zenith–impact differential—successful serve (m)	0.724	0.19	0.61–0.84	0.81
Zenith–impact differential—unsuccessful serve (m)	0.720	0.19	0.61–0.83	0.66
Serve speed—successful serve (km/h)	162.7	8.0	158–167	0.03
Serve speed—unsuccessful serve (km/h)	161.2	7.7	157–166	0.01
Internal Shoulder Rotation—Concentric (210°/s, Nm)	31.9	7.4	27.5–36.4	0.53
External Shoulder Rotation—Concentric (210°/s, Nm)	21.0	4.5	18.3–23.8	0.82
Internal Shoulder Rotation—Eccentric (210°/s, Nm)	38.5	8.7	33–43.7	0.47
External Shoulder Rotation—Eccentric (210°/s, Nm)	32.2	7.9	27.4–37	0.10
Internal Shoulder Rotation—Eccentric (300°/s, Nm)	33.2	6.0	29.6–36.8	0.17
External Shoulder Rotation—Eccentric (300°/s, Nm)	30.1	5.4	26.8–33.4	0.55
Internal Shoulder Rotation—Concentric (300°/s, Nm)	29.0	7.2	24.7–33.3	0.99
External Shoulder Rotation—Concentric (300°/s, Nm)	20.8	4.0	18.4–23.2	1.00

Abbreviations: CI—95% confidence interval; SD—standard deviation; SW—*p* value of Shapiro–Wilk test. Notes: Values represent participant-level means based on the first 10 successful and first 10 unsuccessful serves per player. Within-player SDs describe variability across these 10 trials. Values are mean ± SD. Decimal precision reflects instrument resolution: serve speed 0.1 km/h; kinematic heights and zenith–impact differential 0.001 m; torque 0.1 Nm.

**Table 2 jfmk-10-00438-t002:** Spearman’s coefficients between kinematics variables and isokinetic strength or serve speed in successful serves.

	Zenith—Successful Serve (m)			Impact Height—Successful Serve (m)			Zenith-Impact Differential—Successful Serve (m)		
Variables	ρ	95% CI	*p*	ρ	95% CI	*p*	ρ	95% CI	*p*
IR Shoulder—Concentric (210°/s)	−0.038	−0.58, 0.54	0.901	0.445	−0.19, 0.84	0.128	−0.324	−0.74, 0.38	0.280
ER Shoulder—Concentric (210°/s)	0.198	−0.42, 0.68	0.517	0.456	−0.18, 0.84	0.117	−0.110	−0.66, 0.53	0.721
IR Shoulder—Eccentric (210°/s)	−0.170	−0.67, 0.43	0.578	0.434	−0.22, 0.8	0.138	−0.434	−0.82, 0.22	0.138
ER Shoulder—Eccentric (210°/s)	0.005	−0.55, 0.56	0.986	0.203	−0.42, 0.68	0.505	−0.088	−0.64, 0.53	0.775
IR Shoulder—Eccentric (300°/s)	0.077	−0.50, 0.61	0.803	0.335	−0.30, 0.78	0.263	−0.099	−0.64, 0.5	0.748
ER Shoulder—Eccentric (300°/s)	−0.176	−0.67, 0.43	0.566	0.231	−0.36, 0.7	0.448	−0.396	−0.73, 0.38	0.181
IR Shoulder—Concentric (300°/s)	−0.203	−0.68, 0.42	0.505	0.390	−0.26, 0.80	0.188	−0.434	−0.82, 0.22	0.138
ER Shoulder—Concentric (300°/s)	0.044	−0.54, 0.58	0.887	0.511	−0.12, 0.86	0.074	−0.324	−0.74, 0.38	0.280
Serve speed—successful (km/h)	0.289	−0.31, 0.73	0.338	0.746 ^†^	0.31, 0.93	0.003	−0.217	−0.69, 0.41	0.476

^†^ ρ ≥ 0.69 indicates a statistically detectable effect under 80% power. Abbreviations: ρ—Spearman’s correlation coefficient; CI—95% confidence interval; IR—internal rotation; ER—external rotation. Notes: Spearman’s ρ was applied for bivariate associations. Values represent participant-level correlations (*n* = 13). Confidence intervals were estimated using Fisher’s z transformation. Given the limited sample size and corresponding sensitivity threshold (80% power to detect ρ ≥ 0.69), the results are interpreted as exploratory.

**Table 3 jfmk-10-00438-t003:** Spearman’s coefficients between kinematics variables and isokinetic strength or serve speed in unsuccessful serves.

	Zenith—Unsuccessful Serve			Impact Height—Unsuccessful Serve			Zenith-Impact Differential—Unsuccessful Serve		
Variables	ρ	95% CI	*p*	ρ	95% CI	*p*	ρ	95% CI	*p*
IR shoulder—Concentric (210°/s)	−0.071	−0.60, 0.50	0.817	0.231	−0.37, 0.69	0.448	−0.247	−0.70, 0.35	0.415
ER Shoulder—Concentric (210°/s)	0.148	−0.44, 0.65	0.629	0.324	−0.28, 0.74	0.280	−0.060	−0.59, 0.51	0.845
IR Shoulder—Eccentric (210°/s)	−0.148	−0.63, 0.44	0.629	0.264	0.03, 0.88	0.384	−0.264	−0.71, 0.33	0.384
ER Shoulder—Eccentric (210°/s)	−0.027	−0.57, 0.53	0.929	0.077	−0.50, 0.60	0.803	−0.016	−0.56, 0.54	0.957
IR Shoulder—Eccentric (300°/s)	0.038	−0.54, 0.59	0.901	0.247	−0.35, 0.70	0.415	0.000	−0.56, 0.56	1.000
ER Shoulder—Eccentric (300°/s)	−0.181	−0.67, 0.41	0.553	0.176	0.00, 0.87	0.566	−0.253	−0.71, 0.35	0.405
IR Shoulder—Concentric (300°/s)	−0.236	−0.68, 0.36	0.437	0.209	−0.49, 0.61	0.494	−0.385	−0.83, 0.13	0.194
ER Shoulder—Concentric (300°/s)	−0.011	−0.56, 0.54	0.972	0.374	−0.22, 0.77	0.209	−0.269	−0.71, 0.33	0.374
Serve speed—unsuccessful (km/h)	0.269	−0.33, 0.72	0.375	0.776 ^†^	0.39, 0.93	0.002	−0.030	−0.57, 0.53	0.921

^†^ ρ ≥ 0.69 indicates a statistically detectable effect under 80% power. Abbreviations: ρ—Spearman’s correlation coefficient; CI—95% confidence interval; IR—internal rotation; ER—external rotation. Notes: Spearman’s ρ was applied for bivariate associations. Values represent participant-level correlations (*n* = 13). Confidence intervals were estimated using Fisher’s z transformation. Given the limited sample size and corresponding sensitivity threshold (80% power to detect ρ ≥ 0.69), the results are interpreted as exploratory.

**Table 4 jfmk-10-00438-t004:** Summary of moderation models for successful serves.

Predictor	Moderator	*β*_int	SE	*t*	*p*	95% CI	R^2^
IR Shoulder—Eccentric (210°/s)	Impact height variability	−0.417	0.18	−2.49	0.043	[−0.81, −0.02]	0.551
IR Shoulder—Eccentric (210°/s)	Zenith–impact differential variability	−0.493	0.21	−2.33	0.048	[−0.98, −0.01]	0.533

Note. Standardized coefficients (*β*) reported; *β*_int = X × Z interaction. Two-tailed tests; *n* = 13. Confidence intervals are based on standard OLS estimates. Predictors were standardized; simple slopes were probed at ±1 SD (see Figure 6). R^2^ refers to the full model including main effects (X, Z) and the interaction (X×Z). No bootstrap correction was applied.

## Data Availability

The associated dataset for all performed analyses is available at the Open Science Framework [OSF] repository URL: https://osf.io/f5gdr/?view_only=8d53ca16ed444351841ee8a0edd3e2d9 (accessed on 2 August 2025).

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
