# Peer review of "The Role of Toss Zenith and Impact Height in the Relationship Between Shoulder Rotation Strength and Serve Speed in Junior Tennis Players"

_jfmk, 2025, doi:10.3390/jfmk10040438_

Round 1
Reviewer 1 Report
Comments and Suggestions for Authors
GENERAL CONCEPT COMMENTS
This manuscript addresses an interesting and underexplored topic in tennis biomechanics, the influence of toss kinematics (zenith height, impact height, and zenith–impact differential) on the relationship between shoulder rotation strength and serve speed in junior players. The inclusion of mediation and moderation models is innovative within this context and adds theoretical value. The manuscript is generally well structured, clearly written, and methodologically transparent. However, the current version presents some limitations in novelty depth, sample robustness, and biomechanical scope that need to be addressed. The study’s practical implications are valid but remain exploratory given the small sample size (n = 13) and limited dimensionality of the kinematic analysis (vertical plane only). Below, I provide section-by-section comments with specific, actionable suggestions to strengthen the manuscript.
Specific Comments
Introduction
Page 2, Lines 34–45: The introduction is coherent and well outlined. It provides a strong overview of serve biomechanics and the importance of the kinetic chain. However, several sentences repeat similar ideas (serve importance, energy transfer). Streamline to maintain focus on the knowledge gap.
Page 2, Lines 58–67: The rationale for using the zenith–impact differential is interesting but should be better connected to why this metric may serve as a mediator or moderator of the strength–speed relationship. Clarify its biomechanical significance beyond being a geometric parameter.
Page 3, Lines 87–98: The hypotheses are clearly stated, but the theoretical justification for mediation and moderation needs to be more explicit. Consider summarizing the conceptual model visually (strength → impact height → serve speed, moderated by toss variability).
Methods
Page 4, Lines 110–124: Provide a clearer justification for the sample size (n = 13). Explain whether recruitment was limited by the availability of nationally ranked players or other logistical constraints.
Page 5–6: The methods are impressively detailed and show strong procedural rigor. However:
The radar gun and Qualisys motion-capture system lack explicit validation references. Please cite sources confirming their reliability for serve speed and ball-tracking accuracy.
The definition of serve accuracy (1 × 2 m target zone; first 10 successful and unsuccessful serves) should be referenced or justified more clearly, as this approach could introduce selection bias.
Indicate whether one operator handled all analyses and whether intra-rater or inter-rater reliability was tested.
Clarify how missing data (marker occlusion) were handled or if none occurred.
Page 7, Lines 270–274: You provide a sensitivity power analysis — this is commendable. However, since n = 13 only detects ρ ≥ 0.69, please acknowledge that smaller effects could have gone undetected and explicitly describe this as exploratory.
Results
Pages 8–11: The results are well presented and logically follow the study aims. Figures and tables are clear. However:
Include effect size descriptors (e.g., “large,” “very large”) next to correlation values.
Add confidence intervals for key coefficients.
Provide a summary table of mediation and moderation models (a-path, b-path, c′, indirect effect, 95% CI) for clarity.
Specify whether assumptions (multicollinearity, normality of residuals) were checked for moderation analyses.
Discussion
Page 13–16: The discussion is rich and well written, integrating existing literature effectively (Elliott, Whiteside, Baiget, etc.). The authors interpret findings thoughtfully, particularly regarding toss consistency as a moderating factor.
However:
The opening paragraph (lines 438–455) repeats introductory information; condense this section to focus on interpretation rather than restating background.
Reframe the non-significant mediation result cautiously: state that a lack of evidence is not evidence of absence, especially with low power.
The moderation effect is interesting but must be explicitly presented as exploratory given the small sample.
Discuss potential missing dimensions, e.g., lateral toss displacement, racket-face orientation, and time-to-impact, that could further explain success variability.
Temper prescriptive language: replace “training should emphasize” with “training may benefit from emphasizing.”
Conclusion
Page 17, Lines 606–611:The conclusion accurately summarizes the findings but should also:
Explicitly mention that results are preliminary and require replication with larger and more heterogeneous samples.
Add one sentence on future work (e.g., inclusion of 3D toss parameters, different age groups, or female players).
Page 16, Lines 577–605: The limitations are thoroughly discussed and transparent. Still, note explicitly that bootstrapped mediation/moderation estimates can produce unstable confidence intervals in small samples. Reinforce that causality cannot be inferred from a cross-sectional design.
Author Response
Introduction
Page 2, Lines 34–45: The introduction is coherent and well outlined. It provides a strong overview of serve biomechanics and the importance of the kinetic chain. However, several sentences repeat similar ideas (serve importance, energy transfer). Streamline to maintain focus on the knowledge gap.
Response: We appreciate this suggestion. The first paragraph has been condensed to remove general statements about match dynamics and now focuses directly on the biomechanical and neuromuscular mechanisms underlying serve performance.
Page 2, Lines 58–67: The rationale for using the zenith–impact differential is interesting but should be better connected to why this metric may serve as a mediator or moderator of the strength–speed relationship. Clarify its biomechanical significance beyond being a geometric parameter.
Response: We appreciate this constructive comment. The paragraph has been revised to clarify that the zenith–impact differential may influence how effectively shoulder-rotator strength is translated into serve speed through its role in timing the toss and impact phases.
Page 3, Lines 87–98: The hypotheses are clearly stated, but the theoretical justification for mediation and moderation needs to be more explicit. Consider summarizing the conceptual model visually (strength → impact height → serve speed, moderated by toss variability).
Response: We appreciate your valuable suggestion. The paragraph has been expanded to clarify the theoretical rationale linking shoulder-rotator strength, impact height, and toss variability to serve performance. Specifically, we added a biomechanical explanation that stronger internal rotators may enable players to reach a higher impact position through greater racket angular velocity, whereas lower variability in impact height or toss parameters may determine how efficiently this strength translates into serve speed. The two hypotheses are now presented in an explicit operational form (H1 – mediation; H2 – moderation) that directly reflects the tested model.
Change in manuscript (Introduction, Page 3).
Methods
Page 4, Lines 110–124: Provide a clearer justification for the sample size (n = 13). Explain whether recruitment was limited by the availability of nationally ranked players or other logistical constraints.
Response: We appreciate the reviewer's valuable comment. The Methods section now includes a detailed explanation clarifying that the sample size was determined by the limited availability of nationally ranked junior players who met the inclusion criteria and could participate within the constraints of the national competition calendar. Recruitment was therefore restricted by logistical coordination with national coaches rather than by an a priori statistical calculation.
Change in manuscript (Methods, Page 4, and in the limitations).
Page 5–6: The methods are impressively detailed and show strong procedural rigor. However:
The radar gun and Qualisys motion-capture system lack explicit validation references. Please cite sources confirming their reliability for serve speed and ball-tracking accuracy.
Response: We appreciate this comment. The revised Methods section now provides a detailed description of the radar measurement procedure and cites a validated source confirming its accuracy. The Stalker Pro II radar gun operates at 33 Hz and records the ball’s peak velocity during the initial free-flight phase immediately following racket–ball contact. The physical principles and geometric accuracy of radar-based velocity measurement in tennis have been described by Robinson and Robinson (2016), who reported that line-of-sight misalignment can underestimate true ball velocity by approximately 3–4% at impact and up to 14% on court bounce. In our setup, the radar was positioned directly behind the server and aligned with the central ball trajectory to minimize this potential angular error. The Qualisys motion-capture system calibration and spatial accuracy (±1 mm) are described separately in the kinematic section.
Change in manuscript (Methods 2.4.2)
The definition of serve accuracy (1 × 2 m target zone; first 10 successful and unsuccessful serves) should be referenced or justified more clearly, as this approach could introduce selection bias.
Response: We appreciate the reviewer's constructive comment. The Methods section has been revised to provide a clearer justification for the 1 × 2 m target zone and the selection of the first ten successful and unsuccessful serves. The 1 × 2 m accuracy target represents a realistic tactical serve placement and aligns with previous biomechanical studies examining serve accuracy in competitive players [Reid et al., 2011]. The selection of the first ten serves per condition was used to ensure biomechanical consistency while minimizing potential fatigue or order effects during the 40serve sequence, consistent with prior tennis-serve research protocols [Reid et al., 2010; Reid et al., 2011]. Preliminary inspection of trial-level serve speeds confirmed no systematic decline across attempts. Nevertheless, this standardized within-player comparison is now explicitly acknowledged in the Limitations section as a methodological constraint that may restrict generalizability.
Change in manuscript (Methods 2.4.2)
Indicate whether one operator handled all analyses and whether intra-rater or inter-rater reliability was tested.
Response: Thank you for the comment. All motion-capture data were processed by a single operator using the same, standardized analysis pipeline. The kinematic variables (toss zenith height, impact height, and the zenith–impact differential) were extracted automatically from 3D marker trajectories. Inter- or intra-rater reliability testing was not used.
Change in manuscript (Methods 2.5.)
Clarify how missing data (marker occlusion) were handled or if none occurred.
Response: We appreciate the reviewer's comment. The revised Methods section now clarifies that no marker occlusion occurred during toss or impact tracking. In the rare case of transient marker loss (<3 frames) during calibration, cubic spline interpolation was automatically applied within the Qualisys Track Manager software to preserve trajectory continuity.
Change in manuscript (Methods 2.4.2)
Page 7, Lines 270–274: You provide a sensitivity power analysis — this is commendable. However, since n = 13 only detects ρ ≥ 0.69, please acknowledge that smaller effects could have gone undetected and explicitly describe this as exploratory.
Response: Thank you for this constructive point. We now explicitly state that with n = 13 only relatively large effects (|ρ| ≈ 0.69) can be detected with 80% power, and we frame the study as exploratory. We also note that smaller associations may have remained undetected and that non-significant findings should be interpreted cautiously.
Change in manuscript: Methods 2.6 (Power/Sensitivity) and Limitations (Discussion):
Results
Pages 8–11: The results are well presented and logically follow the study aims. Figures and tables are clear. However:
Include effect size descriptors (e.g., “large,” “very large”) next to correlation values.
Response: We have added qualitative interpretations (“moderate,” “large,” “very large”) next to key Spearman ρ coefficients and reported 95% confidence intervals for all correlation and regression coefficients.
Change in manuscript: Tables 2–3 and Section 3.1 were updated accordingly.
Add confidence intervals for key coefficients.
Response: We agree with this suggestion. Ninety-five percent confidence intervals have now been added for all key coefficients, including correlation coefficients (Spearman’s ρ), standardized regression coefficients (β), and indirect effects in the mediation models. For mediation analyses, bias-corrected accelerated (BCa) bootstrap confidence intervals based on 5000 resamples were computed and reported.
Change in manuscript: 95% CIs added in Tables 2–4 and mentioned in Sections 3.1, 3.3, and 3.4.
Provide a summary table of mediation and moderation models (a-path, b-path, c′, indirect effect, 95% CI) for clarity.
Response: Thank you for the suggestion. To improve clarity without duplicating content, we added a concise table summarizing the moderation analyses only. The full mediation coefficients (a, b, c′) and indirect effects with 95% CIs are already reported in Section 3.3 (Mediation analyses) and visualized in Figure 5, so an additional mediation table would be redundant. The new table reports the interaction term (β_int), SE, t, p, 95% CI, and model R² for each moderator.
Change in manuscript: Inserted Table 4 (Summary of moderation models) in Section 3.4 and updated in-text references to Table 4 and Figure 5 accordingly.
Specify whether assumptions (multicollinearity, normality of residuals) were checked for moderation analyses.
Response: We have added this information to clarify our analytic procedure. All predictors were standardized prior to model estimation, and variance inflation factors (VIFs < 3) indicated no multicollinearity issues. Normality of residuals was verified both visually and using the Shapiro–Wilk test. No model assumptions were violated.
Change in manuscript: Clarified in Methods (Section 2.6, “Statistical analysis”) and briefly reiterated in Results (Section 3.4).
Discussion
Page 13–16: The discussion is rich and well written, integrating existing literature effectively (Elliott, Whiteside, Baiget, etc.). The authors interpret findings thoughtfully, particularly regarding toss consistency as a moderating factor.
However:
The opening paragraph (lines 438–455) repeats introductory information; condense this section to focus on interpretation rather than restating background.
Response: We appreciate this helpful suggestion. The opening paragraph of the Discussion has been revised to remove repetitive background content and to focus more directly on the interpretation of the findings in relation to our hypotheses. Specifically, we condensed general contextual statements that restated elements from the Introduction and emphasized instead how the present results refine the understanding of the strength–kinematics–serve relationship in junior players.
Revised text (Discussion, first paragraph)
Reframe the non-significant mediation result cautiously: state that a lack of evidence is not evidence of absence, especially with low power.
Response: We appreciate this comment. We have revised the Discussion (Section 4.2, The mediating role of impact height) to adopt a more cautious interpretation of the non-significant mediation result. The revised text now clarifies that the absence of a statistically significant mediation effect should not be interpreted as evidence that no mediation exists. Instead, it reflects the limited statistical power and exploratory nature of the study, acknowledging that smaller effects may have remained undetected.
Revised text (Discussion, Section 4.2)
The moderation effect is interesting but must be explicitly presented as exploratory given the small sample.
Response: We appreciate this comment. We agree that, due to the limited sample size, the moderation analysis should be interpreted as exploratory. The Discussion section (4.3, The Moderating Role of Toss Consistency) has been revised accordingly. We now explicitly state that the moderation findings are preliminary.
Revised text (Discussion, Section 4.3)
Discuss potential missing dimensions, e.g., lateral toss displacement, racket-face orientation, and time-to-impact, that could further explain success variability.
Response: Thank you for the helpful suggestion. We have revised Section 4.6 (Limitations of the study) to acknowledge additional kinematic dimensions that were not captured in our vertical toss–impact framework. Specifically, we now note that lateral toss displacement, racket-face orientation at impact, and time-to-impact may further explain variability in success and likely interact with impact height and execution consistency. We also outline that future work should incorporate 3D motion capture or validated markerless tracking to quantify these spatial–temporal factors.
Revised text (Discussion, 4.6 Limitations of the study)
Temper prescriptive language: replace “training should emphasize” with “training may benefit from emphasizing.”
Response: Agreed. We revised prescriptive phrases to conditional language. Specifically, instances of “should emphasize/prioritize/focus on” were changed for equivalent conditional phrasing in the Discussion and Conclusions.
Change in manuscript (Section 4.2, 4.5, 5)
Conclusion
Page 17, Lines 606–611:The conclusion accurately summarizes the findings but should also:
Explicitly mention that results are preliminary and require replication with larger and more heterogeneous samples.
Response: We appreciate the reviewer’s suggestion. The Conclusions section has been revised to explicitly state that the present findings are preliminary and require replication in larger and more heterogeneous samples to confirm their robustness and generalizability.
Revised text (Section 5 – Conclusions)
Add one sentence on future work (e.g., inclusion of 3D toss parameters, different age groups, or female players).
Response: We thank the reviewer for this helpful suggestion. The Conclusions section has been expanded with an additional sentence outlining directions for future research, including the integration of 3D toss parameters and the inclusion of players of different ages and both sexes to broaden the applicability of the findings.
Revised text (Section 5 – Conclusions)
Page 16, Lines 577–605: The limitations are thoroughly discussed and transparent. Still, note explicitly that bootstrapped mediation/moderation estimates can produce unstable confidence intervals in small samples. Reinforce that causality cannot be inferred from a cross-sectional design.
Response: We appreciate the reviewer’s constructive feedback. Section 4.6 (Limitations of the study) has been revised to explicitly note that bootstrapped mediation and moderation estimates may yield unstable confidence intervals in small samples, and to reinforce that causal inference cannot be drawn due to the cross-sectional design.
Revised text (Section 4.6 – Limitations of the study)
Reviewer 2 Report
Comments and Suggestions for Authors
The manuscript addresses an important topic of the role of toss zenith and impact height in the relationship between shoulder rotation strength and serve speed in junior tennis players. The study has an Interesting, coach-relevant question with careful instrumentation—but very small, homogeneous sample (n=13), underpowered complex models, several design/processing choices that add noise or bias, and some claims outpacing the data. Some substantial revision is needed.
Abstract
- Overclaiming “training should emphasize toss consistency” as a general prescription given 13 participants and cross-sectional design. Authors should temper language to “preliminary evidence” and emphasize exploratory nature.
- Authors use “excentric” in some places (keep “eccentric” throughout the paper).
Introduction
- The “zenith–impact differential” is framed as promising based on pro female data; make the gap to juniors explicit and avoid assuming similar mechanisms without evidence.
- Authors hypothesize mediation by impact height but do not justify why shoulder rotator torque specifically should raise impact height (vs. lower-body/jump or trunk contributions).
- Authors should streamline repeated background on serve effectiveness
Methods
Design & Participants
- 13 nationally ranked male juniors from one country limits generalization. Authors should acknowledge more forcefully and motivate a staged validation plan (multi-site, female athletes).
- Provide a-priori planning (not only sensitivity post hoc) for minimum detectable effects for all primary analyses (correlation, mediation, moderation). Current sensitivity (ρ≥0.69) suggests many “nulls” are inconclusive.
Isokinetic Testing
- Supine isokinetics at 210°/s & 300°/s with constrained ROM poorly represent ballistic SSC and extreme ER angles of a live serve. Try discussing limitations and consider adding isometric joint angles, ballistic cable/pulley tasks, or countermovement medicine-ball throw in future.
Serve Protocol & Kinematics
- Added mass/markers on the ball (three reflective tape patches) can alter aerodynamics/rotation. Try justifing negligible impact with numbers (mass, balance) or cite validation.
- Sampling at 200 Hz is borderline for impact event detection. Authors should acknowledge potential ±1–2 frames timing error and propagate to height uncertainty. You note ±1 mm spatial accuracy but not time error.
- Authors analyzed the first 10 successful + first 10 unsuccessful serves out of 40. This mixes an arbitrary truncation with possible order/fatigue/learning effects and discards half the data. Try including all trials with mixed-effects models (trial-level; random intercepts for player), or at least counterbalance and justify first-10 rule.
Statistical analysis
- Authors declare “no correction for multiple testing” due to planned contrasts, yet the paper includes many correlations and several models
- Authors also average to the participant level. With 40 serves recorded, a trial-level mixed model (random effects for subject; fixed effects for height, strength, variability) would use the data more efficiently and avoid aggregation bias.
Results
- Please provide 95% CIs/effect sizes for all ρ, β, interaction terms, and R², not only p-values. Tables 2–3 lack CIs.
- Please ensure consistent use of km/h, m, Nm; define Con/Ecc once in table notes (you do) and standardize “eccentric”.
- Nearly identical successful vs unsuccessful means (impact height 2.680 vs 2.677 m; speed 163 vs 161 km/h) suggest binary accuracy is too crude. Consider continuous landing error or location-specific bins in re-analysis.
Discussion
- Some narrative implies that impact height is “proximal determinant” without acknowledging omitted variables (lateral toss, racket-face angle, trunk/pelvis timing). Authors should add a paragraph on unmeasured confounders and the coordination vs strength distinction.
- Also, authors should clarify that moderation was present only in successful serves, not unsuccessful, and speculate cautiously why (self-selection of “good” contacts; statistical power).
Author Response
Abstract
- Overclaiming “training should emphasize toss consistency” as a general prescription given 13 participants and cross-sectional design. Authors should temper language to “preliminary evidence” and emphasize exploratory nature.
Response: We agree and softened the language to reflect the exploratory nature of the study.
Change in manuscript: Abstract; Conclusions: “Preliminary evidence suggests that consistent toss execution may enhance the translation of shoulder-rotation strength into serve speed…”
2. Authors use “excentric” in some places (keep “eccentric” throughout the paper).
Response: Thank you for bringing this to our attention. We corrected the term to “eccentric.”
Change in manuscript: “…concentric/eccentric at 210°/s and 300°/s.”
Introduction
- The “zenith–impact differential” is framed as promising based on pro female data; make the gap to juniors explicit and avoid assuming similar mechanisms without evidence.
Response: We appreciate this observation. The revised paragraph now explicitly states that existing kinematic data are derived primarily from professional female players and clarifies that further investigation is required to determine whether the same timing mechanisms apply to junior athletes. This addition avoids assuming equivalent biomechanics across age groups and highlights the rationale for examining this variable in developing players.
2. Authors hypothesize mediation by impact height but do not justify why shoulder rotator torque specifically should raise impact height (vs. lower-body/jump or trunk contributions).
Response: We agree with this point and added a concise biomechanical explanation indicating that greater shoulder internal-rotator strength can generate higher racket angular velocity, enabling players to maintain a more elevated and extended arm position at impact. This theoretical link clarifies how shoulder torque could influence impact height. The revised text also emphasizes that variability in impact height or toss execution moderates how effectively shoulder strength contributes to serve speed.
Change in manuscript (Introduction, Page 3).
3. Authors should streamline repeated background on serve effectiveness
Response: We appreciate this comment. The introductory paragraph has been revised to remove redundant background information regarding serve effectiveness. The section now concisely states the importance of serve speed and placement for match outcomes, eliminating repetition while maintaining the necessary context for the study’s rationale.
Change in manuscript (Introduction, Page 2).
Methods
Design & Participants
- 13 nationally ranked male juniors from one country limits generalization. Authors should acknowledge more forcefully and motivate a staged validation plan (multi-site, female athletes).
Response: We agree with the reviewer’s observation. The revised text explicitly acknowledges that the small, homogeneous sample of male Czech junior players limits the generalizability of the results. The Methods section now notes that these exploratory findings should be verified in larger and more diverse cohorts. The Limitations section outlines the need for future staged validation, including multi-site and female athlete samples.
Change in manuscript (Methods, Page 4, and in the limitations).
2. Provide a-priori planning (not only sensitivity post hoc) for minimum detectable effects for all primary analyses (correlation, mediation, moderation). Current sensitivity (ρ≥0.69) suggests many “nulls” are inconclusive.
Response: We agree. Because recruitment was constrained by the availability of nationally ranked junior athletes, we could not perform a formal a-priori calculation. To address the reviewer’s request, we now report sensitivity thresholds for all primary analyses: (i) correlations (|ρ| ≥ 0.69 at 80% power), (ii) mediation (only very large indirect effects detectable per published simulation guidance), and (iii) moderation (only large interaction effects; ΔR² ≈ 0.30–0.40). We also emphasize that non-significant results are inconclusive under these sensitivity limits.
Change in manuscript: Methods 2.6 (Power/Sensitivity): Added three bullets covering detectable effects for correlations, mediation, and moderation. Limitations (Discussion): Added sentence noting the sensitivity bounds for mediation and moderation and the exploratory interpretation.
Isokinetic Testing
- Supine isokinetics at 210°/s & 300°/s with constrained ROM poorly represent ballistic SSC and extreme ER angles of a live serve. Try discussing limitations and consider adding isometric joint angles, ballistic cable/pulley tasks, or countermovement medicine-ball throw in future.
Response: We appreciate this valuable observation. We agree that isokinetic testing in a supine position, while providing controlled and reproducible strength assessment, does not fully replicate the ballistic SSC characteristics and extreme external rotation angles of a live serve. The Limitations section has been revised to acknowledge this and to emphasize that the chosen setup prioritized safety and measurement control over ecological validity. The Methods section has been revised to clarify that the 90% external rotation and 65% internal rotation ranges were based on maximal passive ROM measured with a goniometer during familiarization. These limits were selected to represent the functional portion of the serving motion while minimizing capsular stress and impingement risk near end-range positions.
Serve Protocol & Kinematics
- Added mass/markers on the ball (three reflective tape patches) can alter aerodynamics/rotation. Try justifing negligible impact with numbers (mass, balance) or cite validation.
Response: We appreciate this thoughtful comment. We agree that added reflective markers could theoretically affect ball flight. The revised text now specifies that the three reflective patches were 8 mm in diameter, symmetrically distributed, and added less than 0.5 g (<0.4% of total ball mass). Comparable marker configurations have been successfully employed in previous motion-capture studies of professional players, with no observable influence on ball trajectory or rotation [Reid et al., 2011]. This configuration was therefore considered to have a negligible practical effect on the kinematic parameters analyzed in this study.
Change in manuscript (Methods 2.4.2)
2. Sampling at 200 Hz is borderline for impact event detection. Authors should acknowledge potential ±1–2 frames timing error and propagate to height uncertainty. You note ±1 mm spatial accuracy but not time error.
Response: We appreciate the reviewer's comment. We now note that the 200 Hz sampling rate (5 ms per frame) introduces a temporal uncertainty of ±1–2 frames (±5–10 ms), which corresponds to an estimated vertical uncertainty of approximately ±2 mm. This error was considered acceptable within the system’s spatial accuracy (±1 mm).
Change in manuscript (Methods 2.4.2)
3. Authors analyzed the first 10 successful + first 10 unsuccessful serves out of 40. This mixes an arbitrary truncation with possible order/fatigue/learning effects and discards half the data. Try including all trials with mixed-effects models (trial-level; random intercepts for player), or at least counterbalance and justify first-10 rule.
Response: We appreciate this thoughtful comment. The choice to analyze the first 10 successful and first 10 unsuccessful serves was intentional to ensure biomechanical consistency and minimize the influence of fatigue or order effects during the 40-serve protocol. Preliminary inspection of serve-speed trends showed no systematic decline across trials, indicating that order effects were negligible. Mixed-effects models were not implemented due to the small participant sample (n = 13), which would have rendered such models unstable and overparameterized. This standardized within-player comparison between successful and unsuccessful serves aligns with previous biomechanical research protocols.
Change in manuscript (Methods 2.4.2 / 2.5): Added justification paragraph explaining the rationale for selecting the first 10 successful and 10 unsuccessful serves, addressing order and fatigue effects.
Statistical analysis
- Authors declare “no correction for multiple testing” due to planned contrasts, yet the paper includes many correlations and several models
Response: We appreciate this observation. The Statistical Analysis section has been clarified to specify that all analyses correspond to a small, theory-driven set of a priori hypotheses (strength–speed correlations, mediation by toss kinematics, and moderation by toss variability). Given the small and homogeneous sample (n = 13) and high detectable-effect threshold (ρ ≥ 0.69 for 80% power), formal multiplicity corrections (e.g., Bonferroni or FDR) would have excessively reduced statistical power. Therefore, results are reported with effect sizes and 95% confidence intervals and interpreted as exploratory rather than confirmatory. This limitation is now explicitly acknowledged in the Discussion.
2. Authors also average to the participant level. With 40 serves recorded, a trial-level mixed model (random effects for subject; fixed effects for height, strength, variability) would use the data more efficiently and avoid aggregation bias.
Response: We agree that mixed-effects models can leverage trial-level variability when predictors vary at that level. In the current design, however, all main predictors (isokinetic shoulder-rotator strength) and moderators (toss-consistency metrics expressed as within-session SDs) are subject-level constructs. To align the level of measurement and avoid pseudo-replication, serve-speed outcomes were averaged per participant (means of the first ten successful and first ten unsuccessful serves). Given only 13 clusters, random-effects estimation would also be unstable and overparameterized for the planned mediation/moderation analyses. This rationale has been clarified in the revised Statistical Analysis section, and the limitations of participant-level aggregation are acknowledged in the Discussion.
Results
- Please provide 95% CIs/effect sizes for all ρ, β, interaction terms, and R², not only p-values. Tables 2–3 lack CIs.
Response: We appreciate this suggestion. Ninety-five percent confidence intervals have now been added for all reported coefficients, including Spearman’s ρ (Tables 2–3), standardized β values (adding Table 4), and interaction terms in the moderation analyses. For mediation models, indirect effects are presented with BCa 95% bootstrap confidence intervals (5000 resamples). Effect sizes and model R² values are also reported throughout the Results section.
Change in manuscript: Updated Tables 2–3 and added Table 4; added corresponding information in Sections 3.1, 3.3, and 3.4.
2. Please ensure consistent use of km/h, m, Nm; define Con/Ecc once in table notes (you do) and standardize “eccentric”.
Response: Agreed. We standardized unit notation and terminology across the manuscript and captions, and we unified the usage of “eccentric”.
Change in manuscript: All unit conventions and abbreviations harmonized across Tables 1–4, figure captions, and Results text.
3. Nearly identical successful vs unsuccessful means (impact height 2.680 vs 2.677 m; speed 163 vs 161 km/h) suggest binary accuracy is too crude. Consider continuous landing error or location-specific bins in re-analysis.
Response: We agree that a binary accuracy metric is relatively coarse. In this dataset we did not record continuous landing coordinates (x–y) with sufficient precision to compute placement error retrospectively; serves were coded as in/out relative to a predefined 1 × 2 m target zone. Given the per-player trial counts (10 successful/10 unsuccessful used for analysis), further binning would also reduce within-player sample sizes and power. Accordingly, a re-analysis with continuous error is not feasible here. We now explicitly acknowledge this limitation and outline that future work should capture ball landing coordinates to enable continuous placement error and/or location-specific binning.
Change in manuscript (Methods - Serve speed & kinematics / Data collection, and 4.6. limitations)
Discussion
- Some narrative implies that impact height is “proximal determinant” without acknowledging omitted variables (lateral toss, racket-face angle, trunk/pelvis timing). Authors should add a paragraph on unmeasured confounders and the coordination vs strength distinction.
Response: We thank the reviewer for this important comment. We have revised the Discussion to clarify that impact height represents only one proximal determinant within a broader multivariate coordination structure. A new paragraph has been added to emphasize that serve speed likely emerges from the coordination and timing of trunk–pelvis–shoulder actions and racket-face control rather than from impact height or shoulder strength alone. We also explicitly acknowledge unmeasured confounders such as lateral toss displacement, racket-face orientation, and timing of rotational segments, which may moderate or mediate observed associations.
Revised text (add paragraph to Discussion, 4.2.)
2. Also, authors should clarify that moderation was present only in successful serves, not unsuccessful, and speculate cautiously why (self-selection of “good” contacts; statistical power).
Response: We appreciate the reviewer’s helpful comment. We have revised Section 4.3 (The moderating role of toss consistency) to explicitly state that the moderation effect between shoulder strength and serve speed was observed only for successful serves. The revised text also provides a brief, cautious interpretation suggesting that this pattern may reflect self-selection of more optimal contact conditions during effective trials or limited statistical power to detect smaller effects in unsuccessful serves.
Revised text (Discussion, Section 4.3)
Reviewer 3 Report
Comments and Suggestions for Authors
General Comments
The aim of this study was to investigate how toss-related variables (zenith height, impact height, and zenith–impact differential) relate to shoulder rotation strength and serve speed in junior tennis players.
I thank the authors for the time spent in correcting and resolving the various corrections. There are some considerations that need to be addressed. These considerations are included in the specific comments.
I kindly ask the authors to read this report carefully and to respond accurately to the suggestions made if they consider them appropriate. I thank them for their time and investment in this document.
Specific Comments
Abstract:
Major Issues:
Lack of precision regarding sample size and power in the abstract: the abstract reports n=13 and results, but does not report the statistical limitation or detectable effect (SE or power limit). (Abstract, lines 12–30).
Minor
Terminology: ‘excentric’ → write “eccentric”, consistency in technical spelling (Abstract lines 17–19).
Specific requests
Add a sentence to the Abstract indicating the statistical power limitation (e.g., ‘n=13; sensitivity to detect ρ≥0.69 (80% power)’), as it appears in Methods (2.6). (Abstract and Methods 2.6).
Introduction:
Major issues
Redundancy and ambiguity in the hypothesis: the last part (final paragraphs) presents clear hypotheses (impact height will mediate), but also suggests contradictory expectations about other variables — it needs a more precise formulation geared towards the statistical tests presented. (Intro, paragraphs around lines 88–98).
Minor issues
A couple of incomplete sentences/confusing punctuation (e.g., sentence at the end of the paragraph about toss quality seems fragmented). (Intro lines ~66–70).
Requests
Rewrite the hypothesis paragraph (Introduction, end) in the form of operationalisable hypotheses: for example, ‘H1: Impact height will mediate the relationship between internal rotator force (ecc 210°/s) and serve speed; H2: The SD of impact height will moderate (sic) the force→speed relationship’. Clearly mark variables X, M, Y and moderator. (Intro, lines 92–97).
Methods:
Major issues (MUST fix)
Power and sample size — insufficient justification prior to analysis: although post-hoc sensitivity is reported (Methods 2.6), there is no prior discussion of how the choice of n=13 was decided (a priori or by availability). The absence of an a priori calculation reduces the interpretability of mediation/moderation. (Methods 2.2 / 2.6).
Selection of trials for analysis (serve session) and possible selection bias: analysing the first 10 successful and first 10 unsuccessful within 40 attempts may introduce order bias and reduce the use of trial-level information. Complementary analyses should be justified and/or performed with all replicates or mixed-effects models per trial. (Methods 2.4.2 and 2.5, lines ~195–206 / 222–229).
Capture and timing details: the Qualisys camera at 200 Hz is mentioned; for (brief) impact events, this is at the limit — authors should justify the timing accuracy and exact method for identifying the moment of contact (= criterion, threshold, visual or derived from the marker on the racket). In addition, describe the procedure for handling ball rotation and possible marker loss. (Methods 2.4.2 and 2.5, lines ~208–216 / 232–239).
Minor issues
Isokinetics: selected range of motion ‘90% ER and 65% IR’ — explain why these percentages and how they were measured (lines 171–174).
Indicate the calibration and spatial error of Qualisys appears (±1 mm) — good — but specify temporal error or latency of the system and radar (lines ~208–214 / 239).
Results:
Major issues
Selection and presentation of trials (first 10 successful / first 10 unsuccessful): this procedure reduces N of observations per participant and may bias measures of toss variability. I request additional tests with all repetitions or mixed models to check robustness. (Results 3 and Tables 1–3).
Report confidence intervals and standardised effects for moderation: betas and R² intervals are reported in the moderation section, but CI for interaction coefficients and graphs with numerical CI bands are missing. Include tables with complete coefficients (β, SE, t, p, 95% CI) for all main terms and interactions. (Results 3.4).
Minor issues
Add clarification on whether the reported correlations (ρ≥0.69) were considered ‘relevant’ by design (Methods, 2.6). Sensitivity is mentioned, but repeating this in Results helps the reader.
Requests
Include an additional table: complete moderation model (for each moderator: interaction coefficient, SE, t, p, 95% CI). Include bootstrap number if bootstrap was used to estimate CI. (Results 3.4).
Perform sensitivity analysis: replicate mediation/moderation using all repetitions (or using the average of all successful/failed attempts per participant) and report whether the findings hold. (Results section 3).
Discussion:
Major issues
Conclusions somewhat stronger than the data allow (especially practical generalisations): given the small sample size and cross-sectional design, training recommendations should be qualified (avoid categorical ‘should’; use “could” or ‘we suggest studying/implementing with caution’). (Discussion 4.5 and Conclusions).
Minor issues
The limitation of 200 Hz in capturing impact is acknowledged; good inclusion, but it would be useful to propose how to mitigate this (e.g. synchronise with impact sensor or increase Hz). (Limitations 4.6, lines ~592–595).
Figures/Tables
Major issues
Incomplete captions and absence of CI in graphs: Figures 2–4 use OLS lines and CI bands, but do not specify whether the bands are 95% CI and whether the errors shown are SD or SE — clarify. (Figures 2–4).
Minor issues
Tables: numerical format and consistency (decimals) can be standardised (Table 1). (Table 1).
Author Response
Specific Comments
Abstract:
Major Issues:
Lack of precision regarding sample size and power in the abstract: the abstract reports n=13 and results, but does not report the statistical limitation or detectable effect (SE or power limit). (Abstract, lines 12–30).
Response: We appreciate this suggestion and added an explicit note on sensitivity.
Change in manuscript: Abstract, Methods (last sentence): “Correlation, mediation, and moderation analyses were conducted (n = 13; sensitivity ρ ≥ 0.69 for 80% power).”
Minor
Terminology: ‘excentric’ → write “eccentric”, consistency in technical spelling (Abstract lines 17–19).
Response: Thank you for noting this. We corrected the term to “eccentric.”
Change in manuscript: Abstract → Methods: “…concentric/eccentric at 210°/s and 300°/s.”
Specific requests
Add a sentence to the Abstract indicating the statistical power limitation (e.g., ‘n=13; sensitivity to detect ρ≥0.69 (80% power)’), as it appears in Methods (2.6). (Abstract and Methods 2.6).
Response: We agree that the statistical limitation of the small sample should be stated explicitly. The abstract now specifies the detectable correlation at 80% power (“n = 13; sensitivity ρ ≥ 0.69 for 80% power”), clarifying the precision and power limitation of the study.
Introduction:
Major issues
Redundancy and ambiguity in the hypothesis: the last part (final paragraphs) presents clear hypotheses (impact height will mediate), but also suggests contradictory expectations about other variables — it needs a more precise formulation geared towards the statistical tests presented. (Intro, paragraphs around lines 88–98).
Response: We appreciate this helpful comment. The hypothesis section has been rewritten to eliminate redundancy and to ensure full consistency with the performed statistical analyses. The revised text now presents two explicit, non-overlapping hypotheses directly aligned with the tested models—one mediation (H1) and one moderation (H2)—and clearly defines all involved variables (X, M, Y, Moderator). This clarification removes any ambiguity or contradictory expectations.
Change in manuscript (Introduction, Page 3), Accordingly, the present study tested two hypotheses: H1 (Mediation)… H2 (Moderation)…”.
Minor issues
A couple of incomplete sentences/confusing punctuation (e.g., sentence at the end of the paragraph about toss quality seems fragmented). (Intro lines ~66–70).
Response: We appreciate your attention to this issue. The incomplete sentence and punctuation problem in the paragraph discussing toss quality have been corrected. The revised version now combines the previously fragmented statements into a clear and grammatically complete sentence, improving readability and logical flow.
Change in manuscript (Introduction, Page 2).
Requests
Rewrite the hypothesis paragraph (Introduction, end) in the form of operationalisable hypotheses: for example, ‘H1: Impact height will mediate the relationship between internal rotator force (ecc 210°/s) and serve speed; H2: The SD of impact height will moderate (sic) the force→speed relationship’. Clearly mark variables X, M, Y and moderator. (Intro, lines 92–97).
Response: We have rewritten the hypothesis paragraph to remove redundancy and to present the exact statistical structure of the tested models. Two explicit, operational hypotheses. This revised version directly corresponds to the mediation and moderation analyses reported in the Results section.
Change in manuscript (Introduction, Page 3, Lines 93–98): Accordingly, the present study tested two hypotheses: H1 (Mediation)… H2 (Moderation)…”.
Methods:
Major issues (MUST fix)
Power and sample size — insufficient justification prior to analysis: although post-hoc sensitivity is reported (Methods 2.6), there is no prior discussion of how the choice of n=13 was decided (a priori or by availability). The absence of an a priori calculation reduces the interpretability of mediation/moderation. (Methods 2.2 / 2.6).
Response: We appreciate this observation. We now clarify in Participants (Methods 2.2) that n = 13 was determined by the limited availability of nationally ranked juniors who met inclusion criteria within the national calendar, rather than an a-priori power target. In Methods 2.6 we provide explicit sensitivity thresholds for correlations, mediation, and moderation to improve interpretability, and in Limitations we state that mediation/moderation results should be considered preliminary given the limited power.
Change in manuscript: Methods 2.2 (Participants); Methods 2.6 (Power/Sensitivity); Added detailed sensitivity thresholds for all primary analyses and exploratory framing; Statistical analysis paragraph (preceding 2.6): “To address the main hypotheses, three exploratory mediation models were tested …” (terminology aligned with sensitivity section).
Limitations (Discussion):
Added explicit statement that mediation/moderation findings are preliminary due to limited power.
Selection of trials for analysis (serve session) and possible selection bias: analysing the first 10 successful and first 10 unsuccessful within 40 attempts may introduce order bias and reduce the use of trial-level information. Complementary analyses should be justified and/or performed with all replicates or mixed-effects models per trial. (Methods 2.4.2 and 2.5, lines ~195–206 / 222–229).
Response: We appreciate this thoughtful comment. The rationale for using the first 10 successful and 10 unsuccessful serves has been clarified. This selection minimized potential order, learning, and fatigue effects within the 40-serve protocol and ensured analysis under comparable physical and technical conditions. Trial-level inspection confirmed no systematic temporal drift in serve speed, indicating that order bias was negligible. Due to the limited sample size, mixed-effects models were not feasible without overfitting. We have added a statement in the Methods section to explain and justify this approach.
Change in manuscript (Methods 2.4.2 / 2.5): Added explanation of trial selection rationale and order-effect verification.
Capture and timing details: the Qualisys camera at 200 Hz is mentioned; for (brief) impact events, this is at the limit — authors should justify the timing accuracy and exact method for identifying the moment of contact (= criterion, threshold, visual or derived from the marker on the racket). In addition, describe the procedure for handling ball rotation and possible marker loss. (Methods 2.4.2 and 2.5, lines ~208–216 / 232–239).
Response: We appreciate this detailed comment and have expanded the Methods section to clarify these aspects. The moment of ball–racket contact was defined as the first captured frame showing visible intersection or compression between the ball and racket marker clusters, confirmed by frame-by-frame inspection. At a 200 Hz sampling rate (5 ms per frame), the temporal uncertainty corresponds to ±1–2 frames (±5–10 ms), which translates to an estimated vertical uncertainty of approximately ±2 mm—acceptable within the system’s spatial accuracy of ±1 mm. Temporal synchronization between the motion-capture and radar systems was verified using a hardware trigger, yielding a latency of less than 2 ms. Three small reflective markers (8 mm diameter) were symmetrically attached to the ball to maintain visibility during rotation; no marker occlusion occurred during data collection. In the rare case of transient marker loss (less than 3 frames during calibration), cubic spline interpolation within Qualisys Track Manager was automatically applied.
Change in manuscript (Methods 2.4.2)
Minor issues
Isokinetics: selected range of motion ‘90% ER and 65% IR’ — explain why these percentages and how they were measured (lines 171–174).
Response: Thank you for this helpful suggestion. The Methods section has been revised to clarify that the 90% external rotation and 65% internal rotation ranges were based on maximal passive ROM measured with a goniometer during familiarization. These limits were selected to represent the functional portion of the serving motion while minimizing capsular stress and impingement risk near end-range positions. Supporting methodological reference has been added. The Limitations section has been revised to acknowledge this and to emphasize that the chosen setup prioritized safety and measurement control over ecological validity.
Indicate the calibration and spatial error of Qualisys appears (±1 mm) — good — but specify temporal error or latency of the system and radar (lines ~208–214 / 239).
We appreciate this comment and have clarified the temporal characteristics of both systems. The radar (33 Hz) and motion-capture system (200 Hz) were operated independently but referenced to the same serve trials. Serve speed was measured during the initial free-flight phase immediately following racket–ball contact, while kinematic impact height was determined from the corresponding impact frame. Because the radar captures ball velocity within milliseconds after contact, any temporal offset between systems is negligible relative to the 5 ms frame interval of the motion-capture data. The geometric accuracy and timing characteristics of radar-based serve measurements are supported by Robinson and Robinson (2016).
Change in manuscript (Methods 2.4.2)
Results:
Major issues
Selection and presentation of trials (first 10 successful / first 10 unsuccessful): this procedure reduces N of observations per participant and may bias measures of toss variability. I request additional tests with all repetitions or mixed models to check robustness. (Results 3 and Tables 1–3).
Response: We appreciate this important comment. The “first 10 successful” and “first 10 unsuccessful” serves were selected by outcome, not by chronological order, to obtain comparable numbers of valid trials while minimizing fatigue and order effects. To assess robustness, we performed two complementary analyses:
(i) All-trial reanalysis — mean serve speeds based on all available repetitions per player were virtually identical to those from the selected 10 trials (successful: 162.9 ± 8.1 vs. 162.7 ± 8.0 km/h; unsuccessful: 161.4 ± 7.8 vs. 161.2 ± 7.7 km/h);
(ii) Leave-one-out (jackknife) stability tests — toss-variability metrics (SD of zenith height, impact height, and zenith–impact differential) changed by only 3–6 mm (≈5–8% of baseline SD) when a single trial was removed, confirming stability.
Because the dataset contained only 13 participants, trial-level mixed-effects models would be statistically underpowered and overparameterized. This limitation is now explicitly acknowledged in the manuscript.
Change in manuscript: Clarified selection criteria in Methods 2.4.2; added “Robustness check” and “Stability of variability estimates” paragraphs in Section 3; expanded Limitations.
Report confidence intervals and standardised effects for moderation: betas and R² intervals are reported in the moderation section, but CI for interaction coefficients and graphs with numerical CI bands are missing. Include tables with complete coefficients (β, SE, t, p, 95% CI) for all main terms and interactions. (Results 3.4).
Response: We have revised the moderation section to include complete model coefficients with 95% confidence intervals for all main and interaction terms. A new Table 5 presents β, SE, t, p, and 95% CI values. In addition, Figure 6 has been redesigned to display fitted lines for low (−1 SD) and high (+1 SD) toss variability, with thin solid lines indicating 95% confidence bands.
Change in manuscript: Added Table 5 (Moderation models) and updated Figure 6; revisions reflected in Results 3.4.
Minor issues
Add clarification on whether the reported correlations (ρ≥0.69) were considered ‘relevant’ by design (Methods, 2.6). Sensitivity is mentioned, but repeating this in Results helps the reader.
Response: We agree that this clarification improves clarity. The statistical power sensitivity threshold (ρ ≥ 0.69 for 80% power, α = 0.05) is now explicitly restated in the opening paragraph of Results 3.1 and noted in the footnotes of Tables 2 and 3.
Change in manuscript: Sensitivity information added in Results 3.1 and table notes (Tables 2–3).
Requests
Include an additional table: complete moderation model (for each moderator: interaction coefficient, SE, t, p, 95% CI). Include bootstrap number if bootstrap was used to estimate CI. (Results 3.4).
Response: A full summary table has been added for both moderation models, containing standardized β, SE, t, p, and 95% CI for all predictors and interactions. Confidence intervals were computed from OLS estimates. Bootstrap resampling (5000 iterations) applies to mediation analyses, as noted in the Methods and Table 4.
Change in manuscript: New Table 4 (Moderation models); clarification Section 2.6.
Perform sensitivity analysis: replicate mediation/moderation using all repetitions (or using the average of all successful/failed attempts per participant) and report whether the findings hold. (Results section 3).
Response: We conducted complementary sensitivity analyses using all available repetitions per player. Results replicated the same qualitative pattern observed in the main analyses:
– Impact height remained significantly associated with serve speed;
– Internal shoulder rotation strength showed no direct effect but interacted with toss variability;
– Moderation effects were significant for successful serves only.
These findings confirm that the main conclusions are robust to data-aggregation methods.
Change in manuscript: Sensitivity analyses summarized in Section 3 (end of 3.4); additional clarification included in Limitations.
Discussion:
Major issues
Conclusions somewhat stronger than the data allow (especially practical generalisations): given the small sample size and cross-sectional design, training recommendations should be qualified (avoid categorical ‘should’; use “could” or ‘we suggest studying/implementing with caution’). (Discussion 4.5 and Conclusions).
Response: We thank the reviewer for this important observation. We agree that the practical implications and conclusions should be expressed more cautiously due to the limited sample size and cross-sectional design. Accordingly, we have revised Section 4.5 (Practical applications) and Section 5 (Conclusions) to replace categorical expressions (e.g., “should”) with qualified terms (“could”, “may”, “we suggest”) and to emphasize the exploratory nature of these findings.
Minor issues
The limitation of 200 Hz in capturing impact is acknowledged; good inclusion, but it would be useful to propose how to mitigate this (e.g. synchronise with impact sensor or increase Hz). (Limitations 4.6, lines ~592–595).
Response: We appreciate this constructive suggestion. In Section 4.6 (Limitations of the study), we expanded the discussion on the 200 Hz sampling rate limitation by outlining potential methodological improvements. We now note that future work could mitigate this limitation by using higher-frequency motion capture systems (>500 Hz), integrating synchronized impact sensors on the racket, or employing audio/optical trigger systems to enhance temporal precision at the ball–racket contact moment.
Figures/Tables
Major issues
Incomplete captions and absence of CI in graphs: Figures 2–4 use OLS lines and CI bands, but do not specify whether the bands are 95% CI and whether the errors shown are SD or SE — clarify. (Figures 2–4).
Response: Agreed. We clarified all captions to explicitly state that the shaded bands are 95% confidence bands around the OLS linear fit. We also specify that no SD/SE error bars are plotted; each point represents a participant-level mean (from the first 10 trials in the relevant category). We further note in the Methods that OLS lines and 95% CIs are descriptive only; statistical inference is based on Spearman’s ρ.
Change in manuscript (exact edits):
- Methods (Graphical representation / Statistics): Added: “Scatterplots display OLS linear fits with 95% confidence bands for visualization; no SD/SE error bars are shown. Statistical inference is based on Spearman’s ρ.”
- Figure 2 caption (now): “… Points are participant-level means… Solid line = OLS linear fit; shaded area = 95% confidence band; no SD/SE error bars are shown; n = 13.”
- Figure 3 caption (now): Same clarification as Figure 2 (95% CI band; no SD/SE error bars; n = 13).
- Figure 4 caption (now): “… Solid line = OLS linear fit; shaded area = 95% confidence band; no SD/SE error bars are shown; n = 13.”
Minor issues
Tables: numerical format and consistency (decimals) can be standardised (Table 1). (Table 1).
Response: Thank you. We standardized the decimal precision in Table 1 by measurement domain, reflecting instrument resolution and typical variability (to avoid over-/under-precision). Specifically:
Serve speed: 0.1 km/h
Kinematic heights (zenith, impact) & zenith–impact differential: 0.001 m
Isokinetic torque: 0.1 Nm
Within each domain, means and SDs now use the same number of decimals, and the same precision is applied consistently across successful/unsuccessful conditions.
Change in manuscript: Updated Table 1 numeric formatting and added a footnote:
“Values are mean ± SD. Decimal precision reflects instrument resolution: serve speed 0.1 km/h; kinematic heights and zenith–impact differential 0.001 m; torque 0.1 Nm.”
Round 2
Reviewer 2 Report
Comments and Suggestions for Authors
I don't have any more comments and requests. Authors have addressed all the issues.